# MoFo: Empowering Long-term Time Series Forecasting with Periodic Pattern Modeling

**Jiaming Ma[1], Binwu Wang[1,2,∗], Qihe Huang[1], Guanjun Wang[1]**
**Pengkun Wang[1,2], Zhengyang Zhou[1,2], Yang Wang[1,2,∗]**
[1]University of Science and Technology of China (USTC), Hefei, Anhui, China
[2]Suzhou Institute for Advanced Research, USTC, Suzhou, Jiangsu, China
{JiamingMa, hqh, always}@mail.ustc.edu.cn
{wbw2024, pengkun, zzy0929, angyan}@ustc.edu.cn

## Abstract

The stable periodic patterns present in the time series data serve as the foundation for long-term forecasting. However, existing models suffer from limitations such as continuous and chaotic input partitioning, as well as weak inductive biases, which restrict their ability to capture such recurring structures. In this paper, we propose MoFo, which interprets periodicity as both the correlation of period-aligned time steps and the trend of period-offset time steps. We first design period-structured patches—2D tensors generated through discrete sampling—where each row contains only period-aligned time steps, enabling direct modeling of periodic correlations. Period-offset time steps within a period are aligned in columns. To capture trends across these offset time steps, we introduce a period-aware modulator. This modulator introduces an adaptive strong inductive bias through a regulated relaxation function, encouraging the model to generate attention coefficients that align with periodic trends. This function is end-to-end trainable, enabling the model to adaptively capture the distinct periodic patterns across diverse datasets. Extensive empirical results on widely used benchmark datasets demonstrate that MoFo achieves competitive performance while maintaining high memory efficiency and fast training speed. Our code is available at official repository .

## 1 Introduction

Long-term time series forecasting (LTSF) has found widespread applications across various domains [10, 12, 13, 36, 62, 78, 79], with its core challenge lying in understanding and modeling the inherent periodic patterns present within data [20]. To address this, various cutting-edge models have been proposed, among which Transformer [43] has emerged as the de facto backbone for capturing long-range dependencies in LTSF tasks [6, 24, 62, 72]. Despite the promising results achieved, we identify two underexplored potentials that remain to be fully harnessed.

❶ *Continuous but Chaotic Time Steps of Input*. Existing models often input consecutive time steps, and popular patch-based methods are no exception. Specifically, the patching method partitions the input sequence in a continuous manner using either convolutional down sampling [22] or sliding window strategies [5, 32, 75], which we refer to as the continuous patch. Taking the Electricity dataset as an example, as shown in the red box in Figure 1(a), a consecutive patch may contain both time steps that are phase-aligned across each period (referred to as period-aligned) and those that are misaligned within each period (period-offset). We further visualize the pairwise correlations between

---

∗Binwu Wang and Yang Wang are corresponding authors.

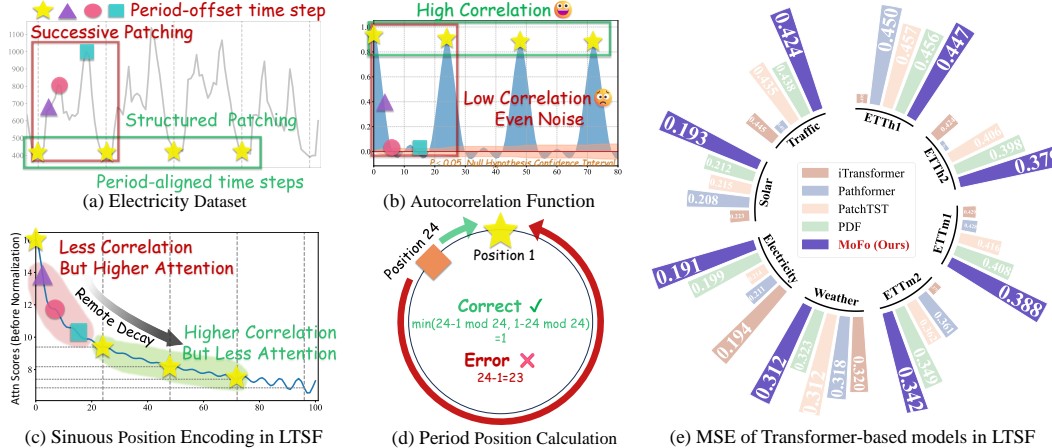

Figure 1: Case visualization and analysis on real-world datasets. (a) Continuous patching strategy. (b) Correlation coefficients within patches. The orange region represents the 95% confidence interval for the null hypothesis from Bartlett's Test [2]. Correlation coefficients that fall within this interval are not statistically significant and cannot be rejected as noise [4]. (c) Attention weights exhibit remote attenuation due to sinuous positional encoding. (d) Absolute and relative position distance in the period perspective. (e) Performance of Transformer-based models for LTSF with 720 horizon.

the first time step in the patch and subsequent ones in Figure 1(b). It is evident that period-aligned time steps, such as 8 AM every day, exhibit strong correlations. In contrast, period-offset time steps rapidly decaying correlations and may even introduce noise. Therefore, the internal correlations within a continuous patch are chaotic, which hampers the model's ability to learn periodic patterns.

❷ *Weak Inductive Bias for Periodicity.* Due to the permutation invariance of the self-attention mechanism, sinusoidal positional encodings [16, 32] or timestamp position embeddings [62, 77, 79] are commonly used to inject temporal position information. However, as shown in Figure 1(c), sinusoidal encodings may cause attention weights to decay gradually with increasing position distance [50], which conflicts with the repetitive nature of periodic signals. In contrast, timestamp position embeddings encode absolute time intervals, leading to large distances between strongly correlated and phase-aligned time steps. For example, in a daily-period electricity dataset, 8 AM on Monday and 7 AM on Tuesday may exhibit high correlation due to their similar daily phases. However, as shown in Figure 1(d), their absolute time difference is encoded as 23, whereas a more reasonable alternative — their relative cyclical distance — is only 1. Large temporal distances may mislead the model into assigning diminished attention weights. Such weak inductive bias toward periodicity in Transformer further hampers the ability to capture periodic patterns from consecutive and chaotic input.

To address these limitations, we propose MoFo, a novel Trans**Fo**rmer architecture with **Mo**dulator that explicitly models periodic patterns by capturing both the correlations among period-aligned time steps and the trends across period-offset ones. Our approach introduces two key components: Period-structured Patching and the Period-Aware Modulator. The former discretely samples time steps to arrange the input into a 2D tensor where each row represents a patch comprising only period-aligned time steps. Period-offset time steps within the period are realigned across columns (i.e., across patches). By modeling the features within each patch, the model can directly learn the underlying periodic correlations. To capture cross-patch periodic trends, we introduce the Period-Aware Modulator, which generates an attention modulation term through a regulated regulated relaxation function. This function assigns attention coefficients based on the periodic relative distances between time steps, encouraging the resulting attention scores to align with the underlying periodic patterns. As a result, the attention mechanism is infused with strong inductive biases that favor periodic dependencies. Crucially, this function is end-to-end trainable, enabling it to dynamically adapt the modulation behavior to the specific periodic characteristics of the input data.

Our contributions are fourfold: ❶ We propose a novel perspective for periodic pattern modeling, termed MoFo, which integrates two innovative strategies for explicit periodicity-aware time series modeling. ❷ We design a Period-Structured Patching strategy that separately manages period-

aligned and period-offset time steps, enabling the model to directly capture both periodic correlations and trends. ❸ We introduce a Period-Aware Modulator, which encourages the Transformer to generate attention coefficients that align with underlying periodic trends through a learnable regulated relaxation function. ❹ Extensive experiments demonstrate that our model achieves competitive forecasting performance while significantly improving computational efficiency — offering up to $14\times$ memory savings and $10\times$ faster training speed compared to state-of-the-art methods.

## 2 Related Work

In recent years, deep learning methods have achieved remarkable success across a broad spectrum of time series tasks, such as forecasting [17, 26, 31, 54, 64, 65, 67, 71], imputation [52], classification [25, 58], and anomaly detection [35, 66, 70]. Among these, Long-Term Series Forecasting (LTSF) has attracted especially intense interest owing to its foundational role in both academic research and practical applications [30, 51, 55]. Existing deep learning approaches for LTSF can be broadly grouped into four categories according to how they handle the temporal dimension: RNN-based, TCN-based, MLP-based (discussed in Appendix B), and Transformer-based models. The Transformer architecture has emerged as the prevailing choice for LTSF, primarily because of its strong ability to capture long-range temporal dependencies. Recent studies have largely concentrated on enhancing both its computational efficiency and modeling power [28, 29], with significant advances driven by two core elements: the self-attention mechanism and patch-based modeling strategies.

**Variants of Self-Attention Mechanism for Time Series Forecasting.** One of the most interesting aspects of Transformer in LTSF is its self-attention mechanism. LogTrans [16] introducing a convolutional LogSparse attention that ensured long-distance interactions while reducing the number of interactions, is an early and influential attempt to apply Transformer. Informer [77] proposes ProbSparse attention while combining with a distillation mechanism to select the most representative query vectors to compute the attention scores. Autoformer [62] utilizes a decomposition architecture to discover dependencies for building sequence-level connections based on their aggregation of similar subsequences. FEDformer [79] leverages the attention mechanism in the frequency domain, providing the capture of the underlying oscillatory modes and their intensities. Despite the efficacy of these methodologies in reducing computational expense, they often encounter information bottlenecks in long sequence input [33].

**Patch-based Time Series Forecasting Method.** Patch-based approaches are widely adopted for efficient time series representation, where the input sequence is divided into consecutive segments (patches). This strategy not only enhances the computational efficiency of the Transformer backbone but also promotes more effective modeling of localized temporal dynamics within each patch [11, 60]. PatchTST [32] explicitly reduces the length of input sequence and preserves the local semantic information by dividing the time series into smaller subsequences. Pyraformer [22] employs a pyra-midal attention mechanism, wherein the input sequence is downsampled into patches by multilayer convolution, allowing attention to be applied at coarse scales. Crossformer [75] proposes two-stage attention with a dynamic routing mechanism that performs the patch operation from the sequence dimensionality and the channel dimensionality. Pathformer [5] selects multiple patch volumes at the same time and uses a multi-scale router to determine the interactions between different patches to weaken the impact of the selection of patch volume. All existing patch strategies are sequential along the temporal dimensionality, which does not facilitate effective capture and learning of periodic patterns [11, 60].

## 3 MoFo

Given $\mathbf{X} = [\mathbf{x}_1, \mathbf{x}_2, \ldots, \mathbf{x}_T]^\top \in \mathbb{R}^T$ with the look-back window $T$ where $\mathbf{x}_t \in \mathbb{R}$ represents the time series data on time step $t$, the objective of LTSF is to forecast the next $L$ values $\mathbf{Y} = [\mathbf{x}_{T+1}, \mathbf{x}_{T+2}, \ldots, \mathbf{x}_{T+L}]^\top \in \mathbb{R}^L$. Effective long-term forecasting hinges on the ability to model the intrinsic periodic patterns underlying the data. In this work, as shown in Figure 2, we propose MoFo which integrates two key components: Period-structured Patch and the Period-Aware Modulator, which jointly enhance the model's capacity to capture and exploit periodic patterns.

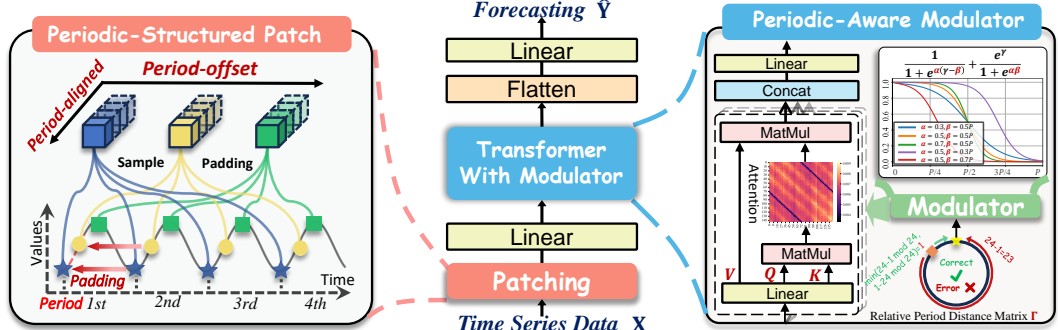

Figure 2: The details of MoFo which contains two core contributions for accurate modeling of periodic patterns: Period-structured Patch and Period-Aware Modulator.

## 3.1 Period-structured Patch in MoFo

Patch-based input representation method has become a popular approach to improve the efficiency of long-time-series modeling of Transformer. However, as discussed in the introduction section, the pairwise correlations between time steps within a continuously partitioned patch can be weak or even negative, which may undermine the model's ability to capture periodic patterns. To address this challenge, we propose a Period-structured Patching strategy. This approach discretely samples period-aligned time steps from the input sequence to form patches. Specifically, given a period length $P$, which is typically much smaller than the input sequence length $T$ (i.e., $P \ll T$) in LTSF settings, we compute $N_P = \lceil T/P \rceil$ as the number of period-length segments required to cover the input time series $\mathbf{X}$. Since the length $T$ of the input time series may not be an integer multiple of the period length $P$, it becomes necessary to apply padding so that all time segments have uniform length without discarding the trailing portion of the input.

**Input Padding.** Our proposed padding strategy fills incomplete periodic segments with data from adjacent periods. Specifically, we start from the current time step and move backward to delineate periods of length $P$, and if necessary, we prepend the input time series with the first $P - (T \bmod P)$ time steps of the first complete period, as shown in Figure 2. This ensures a seamless continuation of the sequence while retaining its underlying periodic structure. As a result, the input series is extended to $\mathbf{X}_{pad} \in \mathbb{R}^{T'}$ with $T'=P*\lceil T/P \rceil$, and the padded series can be formally expressed as:

$$\mathbf{X}_{pad} = \begin{cases} \text{Concat}\left(\mathbf{X}_{(T \bmod P):P}, \mathbf{X}\right), & \text{if } T \bmod P > 0, \\ \mathbf{X}, & \text{if } T \bmod P = 0. \end{cases} \quad (1)$$

**Sampling Patch and Unflatten.** We sample time steps at periodic intervals (i.e., period-aligned time steps) from $\mathbf{X}_{pad}$ and group them into the same patch. For example, for $i$-th patch, it can be denoted as $\overline{\mathbf{X}}^i = [x_i, x_{i+P}, \cdots, x_{i+P*\lceil T/P \rceil}] \in \mathbb{R}^{\lceil T/P \rceil}$, where $x_{i+P}$ means the data point at the time step $(i + P)$ in $\mathbf{X}_{pad}$. Then we unflatten $\overline{\mathbf{X}}$ to generate the patch-structure input $\mathbf{X}_{in}$,

$$\mathbf{X}_{in} = \text{Unflatten}\left(\overline{\mathbf{X}}\right) \in \mathbb{R}^{P \times \lceil T/P \rceil}. \quad (2)$$

where $P$ is the number of patches (also equal to the period length). As illustrated in Figure 2, the structured input $\mathbf{X}_{in}$ possesses two key properties: ❶ Each row (within a patch) contains $\lceil T/P \rceil$ time steps, with all steps period-aligned within their respective periods. This alignment establishes structured temporal dependencies across periods, enabling the explicit modeling of long-range periodic dependencies. ❷ Each column (across patches) contains $P$ time steps from a complete period. By capturing correlations among these patches, the model can effectively learn the underlying periodic trends across period-offset time steps.

**Periodic Dependency Modeling.** Due to this desirable property, even a simple neural architecture—such as a single-layer MLP—can suffice for capturing the underlying periodic dependency. The design is formally described as follows:

$$\mathbf{Z} = \mathbf{X}_{in}\mathbf{W}_{in} + \mathbf{b}_{in} \in \mathbb{R}^{P \times d}, \quad (3)$$

where $\mathbf{W}_{in} \in \mathbb{R}^{\lceil T/P \rceil \times d}$ and $\mathbf{b}_{in} \in \mathbb{R}^d$ are learnable parameters. And $\mathbf{Z}$ is the corresponding output.

## 3.2 Period-Aware Modulator for Periodic Trend Modeling

MoFo further incorporates an enhanced Transformer to model the periodic trends across period-offset time steps within a periods. To further strengthen the model's representational capability, we first modify the standard Transformer architecture (details in Appendix C). Moreover, we introduce a Period-Aware Modulator, which integrates strong inductive biases to address the permutation invariance of self-attention, thereby enhancing the model's ability to capture period trends.

### 3.2.1 Period-Aware Modulator

When computing the attention scores, we explicitly introduce a strong inductive bias: the attention generated by the Transformer is encouraged to align with periodic trends. We achieve this by incorporating a modulation term into the attention computation. First, we design periodic positional encodings that effectively capture the relative distances within a period — a key distinction from existing approaches such as sinusoidal or timestamp-based positional encodings. Specifically, the relative distance between the $i$-th period and the $j$-th period is computed as follows,

$$\gamma_{ij} = \min\{(i-j) \bmod P, (j-i) \bmod P\} \in [0, \lfloor P/2 \rfloor], \tag{4}$$

this strategy ensures that time steps that are periodically close remain proximal. For example, in a traffic dataset with an hourly sampling frequency and a daily period (i.e., 24 time steps), the relative period distance between 8:00 AM on Monday and 7:00 AM on Tuesday is 1, while their absolute distance is 23. This allows the model to better perceive and learn periodic trends. and we can get a relative periodic distance matrix, which is denoted as $\mathbf{\Gamma} = \{\gamma_{ij}\}_{i,j=1}^{P} \in \mathbb{R}^{P \times P}$.

We proceed by introducing a modulation term within the attention computation, designed to encourage the model to prioritize temporal dependencies that align with the underlying cyclical trend of the data. Specifically, we initially define a modulation matrix $\mathbf{M} \in \{0,1\}^{P \times P}$, where each entry $\mathbf{M}_{ij}$ is an indicator that reveals whether time steps i and j are close in terms of cyclical distance. A natural way to construct this matrix is by using the Heaviside Step function $\mathcal{H} : \mathbb{R} \to \{0,1\}$: $\mathbf{M}_{ij} = \mathcal{H}(\beta - \gamma_{ij})$, and its logarithmic value can be defined as:

$$\mathbf{M}_{ij} = \begin{cases} 1, & \text{if } \gamma_{ij} \leq \beta, \\ 0, & \text{if } \gamma_{ij} > \beta, \end{cases} \iff \log \mathbf{M}_{ij} = \begin{cases} 0, & \text{if } \gamma_{ij} \leq \beta, \\ -\infty, & \text{if } \gamma_{ij} > \beta. \end{cases} \tag{5}$$

where $\gamma_{ij}$ is the periodic distance between $i$-th period and $j$-th period calculated above. And $\beta \geq 0$ is a distance penalty threshold controlling the penalization distance. Hence, the attention coefficient between the $i$-th patch and the $j$-th patch is satisfying,

$$\exp\left(\frac{\mathbf{Q}_i \mathbf{K}_j^\top}{\sqrt{d_h}} + \log \mathbf{M}_{ij}\right) = \begin{cases} \exp\left(\frac{\mathbf{Q}_i \mathbf{K}_j^\top}{\sqrt{d_h}}\right), & \text{if } \gamma_{ij} \leq \beta, \\ 0, & \text{if } \gamma_{ij} > \beta. \end{cases} \tag{6}$$

The attention score is considered valid only if the cyclical distance between time step $j$ and $i$ is no more than $\beta$. Otherwise, the attention coefficient is penalized due to the distance and set to zero. Finally, our attention mechanism can be written as,

$$\text{Attention}(\mathbf{Q}, \mathbf{K}, \mathbf{V}) = \text{Softmax}\left(\frac{\mathbf{Q}\mathbf{K}^\top}{\sqrt{d_h}} + \log \mathbf{M}\right)\mathbf{V}, \tag{7}$$

$$\text{where} \quad \mathbf{Q} = \mathbf{Z}\mathbf{W}_Q^j \in \mathbb{R}^{P \times d_h}, \mathbf{K} = \mathbf{Z}\mathbf{W}_K^j \in \mathbb{R}^{P \times d_h}, \mathbf{V} = \mathbf{Z}\mathbf{W}_V^j \in \mathbb{R}^{P \times d_h}, \tag{8}$$

where $\mathbf{W}_Q^j$, $\mathbf{W}_K^j$, and $\mathbf{W}_V^j \in \mathbb{R}^{d \times d_h}$ are learnable parameters.

### 3.2.2 Regulated Relaxation Function for Smooth Approximation

There are still two points with the potential for improvement: ❶ Due to the non-differentiability of the discrete truncation operation in the step function, $\beta$ cannot be optimized through backpropagation, which limits the flexibility and adaptability of the method. ❷ Although the modulated attention mechanism implicitly incorporates temporal positional information by preserving attention coefficients only between time steps with small periodic distances, it imposes uniform inductive biases on nearby steps. As a result, the model still faces challenges related to permutation invariance [15].

To address these limitations, we proposed a regulated relaxation function to approximate the Heaviside Step function to generate the modulator. In contrast to the sharp Heaviside Step function, our function $\mathcal{S}$ exhibits a smooth attenuation trend, as shown in Figure 2, which is defined as follows:

---

**Theorem 1. Regulated Relaxation Function**

Define a continuous differentiable function $\mathcal{S}(\cdot; \alpha, \beta) : \mathbb{R}^+ \cup \{0\} \to [0, 1]$ as follows,

$$\mathcal{S}(\gamma; \alpha, \beta) = \frac{1}{1 + \exp(\alpha(\gamma - \beta))} + \frac{\exp(-\gamma)}{1 + \exp(\alpha\beta)} \in [0, 1]. \tag{9}$$

where the regulated parameter $\alpha > 0$ control the gradient of attenuation and $\beta > 0$ is the distance penalty threshold. This function has following properties:
(1) $\mathcal{S}(\gamma; \alpha, \beta)$ is the smooth approximation of $\mathcal{H}(\beta - \gamma_{ij})$ for arbitrary $\gamma \geq 0$ satisfies,

$$\mathcal{S}(0; \alpha, \beta) = 1, \quad \mathcal{S}(+\infty; \alpha, \beta) = 0, \quad \forall \alpha, \beta > 0. \tag{10}$$

(2) The cumulative error upper bound of this smooth approximation satisfies,

$$\int_0^{+\infty} |\mathcal{H}(\beta - \gamma) - \mathcal{S}(\gamma; \alpha, \beta)| \, \mathrm{d}\gamma < \frac{2\log 2}{\alpha} + \frac{1}{1 + \exp\alpha} \to 0^+ \quad (\alpha \to +\infty). \tag{11}$$

---

where the proof of Theorem 1 is provided in Appendix A.1. Regulated relaxation function takes relative periodic distance matrix $\mathbf{\Gamma} = \{\gamma_{ij}\}_{i,j=1}^P \in \mathbb{R}^{P \times P}$ to generate the attention modulation term. Finally, the formula of our attention with **R**egulated **R**elaxation **F**unction (RRF) is as follows,

$$\mathrm{RRF}(\mathbf{Q}, \mathbf{K}, \mathbf{V}) = \mathrm{Softmax}\left(\frac{\mathbf{Q}\mathbf{K}^\top}{\sqrt{d_h}} + \log \mathcal{S}(\mathbf{\Gamma}; \alpha, \beta)\right)\mathbf{V} \in \mathbb{R}^{P \times d_h}, \tag{12}$$

$$\text{where} \quad \mathbf{Q} = \mathbf{Z}\mathbf{W}_Q^j \in \mathbb{R}^{P \times d_h}, \mathbf{K} = \mathbf{Z}\mathbf{W}_K^j \in \mathbb{R}^{P \times d_h}, \mathbf{V} = \mathbf{Z}\mathbf{W}_V^j \in \mathbb{R}^{P \times d_h}, \tag{13}$$

where $\alpha$ and $\beta$ are learnable parameters, and $\mathbf{Q}, \mathbf{K},$ and $\mathbf{V}$ are the linear projections of $\mathbf{Z}$.

**Advantages for Periodic Modeling.** ❶ The key distance penalty threshold $\beta$ is defined as a learnable parameter, which can be adaptively learned from the data. This enables more accurate and dynamic modeling of periodic trend. ❷ The modulation term varies smoothly with the periodic relative distance, implicitly encoding discriminative positional information of time steps. This effectively addresses the permutation-invariant limitation inherent in the standard attention mechanism. ❸ The function exhibits a smoother trend and can be flexibly controlled through the learnable parameter $\alpha$. This adaptability allows our Transformer architecture to customize personalized modulation behaviors for time series with diverse periodic characteristics, thereby refining the attention process to better capture periodic dependencies.

## 3.3 Forecasting and Optimization

The Final time series prediction values is obtained by summing the flattened and linearly transformed outputs of the Transformer backbone $\vec{\mathbf{Z}} \in \mathbb{R}^{P \times d}$ as follows:

$$\widehat{\mathbf{Y}} = \mathrm{Flatten}(\vec{\mathbf{Z}})\mathbf{W}_{out} + \mathbf{b}_{out} \in \mathbb{R}^L, \tag{14}$$

where $\mathbf{W}_{out} \in \mathbb{R}^{(P*d) \times L}$ and $\mathbf{b}_{out} \in \mathbb{R}^L$ are learnable parameters. When the time series involves multiple variables (i.e., $C > 1$), we adopt channel-independent learning and compute the relative loss weights based on the maximum loss across channels, dynamically adjusting the loss weights during training to promote equal learning with stable convergence. This process is formulated as follows:

$$\mathcal{L}^* = \omega \sum_{c=1}^C \frac{\mathcal{L}(\mathbf{Y}_c, \hat{\mathbf{Y}}_c)}{\|\mathcal{L}(\mathbf{Y}_c, \hat{\mathbf{Y}}_c)\|}, \quad \omega = \max\left\{\|\mathcal{L}(\mathbf{Y}_c, \hat{\mathbf{Y}}_c)\|; c \in \{1, 2, \ldots, C\}\right\}, \tag{15}$$

where $\mathbf{Y}_c, \hat{\mathbf{Y}}_c$ are the ground-truth values and prediction values of the channel $c$.

**Complexity Analysis.** MoFo's computational complexity has a quadratic dependence on period length but is independent of the input sequence length, making its efficiency highly favorable.

For example, in daily-period datasets with hourly sampling (e.g., 24 time steps per period), the computational cost is minimal relative to the total sequence length in long-term forecasting tasks, such as 720 time steps.

# 4 Experiment

## 4.1 Experimental Setup

**Datasets.** We conduct our experiments on widely used real-world time series datasets with periodic pattern from 4 different domains, including ETTh1, ETTh2, ETTm1, ETTm2, Weather, Solar Energy, Electricity, and Traffic. A summary of all datasets is provided in Table 1.

Table 1: Statistics of used datasets.

| Dataset | ETTh1 | ETTh2 | ETTm1 | ETTm2 | Weather | Solar Energy | Electricity | Traffic |
|---|---|---|---|---|---|---|---|---|
| # Channels | 7 | 7 | 7 | 7 | 21 | 137 | 321 | 862 |
| # Samples | 14,400 | 14,400 | 57,600 | 57,600 | 52,696 | 52,560 | 26,304 | 17,544 |
| Frequency | 1 hour | 1 hour | 15 mins | 15 mins | 10 mins | 10 mins | 1 hour | 1 hour |
| Split ratio | 6:2:2 | 6:2:2 | 6:2:2 | 6:2:2 | 7:1:2 | 6:2:2 | 7:1:2 | 7:1:2 |

**Settings.** Our experiments are conducted on an NVIDIA A100 GPU with 40GB memory, using PyTorch under Python 3.11.5. We implement our method within the TFB platform [34] to ensure a fair comparison. Following the evaluation protocol in TFB [34], we report the best performance achieved over look-back window lengths $T \in \{96, 336, 512\}$ and forecasting horizons $L \in \{96, 192, 336, 720\}$. Model performance is evaluated using two standard metrics: mean squared error (MSE) and mean absolute error (MAE). To maintain fairness in evaluation, we disable the "Drop Last" batch-sampling trick[18]. We use the Adam optimizer [14] with the $L_1$ loss function from the FreDF strategy [53].

Table 2: Performance comparisons for LTSF. The **best** and second best are marked in corresponding colors. All experimental results are selected from the best performance under the look-back window length $T \in \{96, 336, 512\}$.

| Data | Metric | MoFo (Ours) MSE | MAE | DUET (2025) MSE | MAE | PDF (2024) MSE | MAE | iTransformer (2024) MSE | MAE | Pathformer (2024) MSE | MAE | CycleNet (2024) MSE | MAE | TimeMixer (2024) MSE | MAE | PatchTST (2023) MSE | MAE | Crossformer (2023) MSE | MAE | DLinear (2023) MSE | MAE |
|---|---|---|---|---|---|---|---|---|---|---|---|---|---|---|---|---|---|---|---|---|---|
| ETTh1 | 96 | 0.360 | 0.389 | **0.352** | **0.384** | 0.360 | 0.391 | 0.386 | 0.405 | 0.372 | 0.392 | 0.372 | 0.394 | 0.372 | 0.401 | 0.377 | 0.397 | 0.411 | 0.435 | 0.379 | 0.403 |
| | 192 | 0.397 | 0.413 | 0.398 | 0.409 | 0.392 | 0.414 | 0.424 | 0.440 | 0.408 | 0.415 | 0.413 | 0.430 | 0.423 | 0.437 | 0.409 | 0.425 | 0.409 | 0.438 | 0.408 | 0.419 |
| | 336 | **0.407** | **0.424** | 0.414 | 0.426 | 0.418 | 0.435 | 0.449 | 0.460 | 0.438 | 0.434 | 0.430 | 0.429 | 0.438 | 0.450 | 0.431 | 0.444 | 0.433 | 0.457 | 0.440 | 0.440 |
| | 720 | 0.447 | 0.454 | 0.429 | 0.455 | 0.456 | 0.462 | 0.495 | 0.487 | 0.450 | 0.463 | 0.448 | 0.464 | 0.486 | 0.484 | 0.457 | 0.477 | 0.501 | 0.514 | 0.471 | 0.493 |
| ETTh2 | 96 | 0.273 | 0.334 | 0.270 | 0.336 | 0.276 | 0.341 | 0.297 | 0.348 | 0.279 | 0.336 | 0.277 | 0.341 | 0.281 | 0.351 | 0.274 | 0.337 | 0.728 | 0.603 | 0.300 | 0.364 |
| | 192 | 0.327 | 0.373 | 0.332 | 0.374 | 0.339 | 0.382 | 0.372 | 0.403 | 0.345 | 0.380 | 0.341 | 0.385 | 0.349 | 0.387 | 0.348 | 0.384 | 0.723 | 0.607 | 0.387 | 0.423 |
| | 336 | 0.361 | 0.405 | 0.353 | 0.397 | 0.374 | 0.406 | 0.388 | 0.417 | 0.378 | 0.408 | 0.370 | 0.411 | 0.366 | 0.413 | 0.377 | 0.416 | 0.740 | 0.628 | 0.490 | 0.487 |
| | 720 | 0.379 | 0.425 | 0.382 | 0.425 | 0.398 | 0.433 | 0.424 | 0.444 | 0.437 | 0.455 | 0.424 | 0.451 | 0.401 | 0.436 | 0.406 | 0.441 | 1.386 | 0.882 | 0.704 | 0.597 |
| ETTm1 | 96 | 0.286 | 0.335 | 0.279 | 0.333 | 0.286 | 0.340 | 0.300 | 0.353 | 0.290 | 0.335 | 0.297 | 0.344 | 0.293 | 0.345 | 0.289 | 0.343 | 0.314 | 0.367 | 0.300 | 0.345 |
| | 192 | 0.320 | 0.363 | 0.320 | 0.358 | 0.321 | 0.364 | 0.341 | 0.380 | 0.337 | 0.363 | 0.332 | 0.365 | 0.335 | 0.372 | 0.329 | 0.368 | 0.374 | 0.410 | 0.336 | 0.366 |
| | 336 | 0.347 | 0.382 | 0.348 | 0.377 | 0.354 | 0.383 | 0.374 | 0.396 | 0.374 | 0.384 | 0.366 | 0.386 | 0.365 | 0.386 | 0.362 | 0.390 | 0.413 | 0.432 | 0.367 | 0.386 |
| | 720 | 0.388 | 0.411 | 0.405 | 0.408 | 0.408 | 0.415 | 0.429 | 0.430 | 0.428 | 0.416 | 0.417 | 0.414 | 0.426 | 0.412 | 0.413 | 0.423 | 0.753 | 0.613 | 0.419 | 0.416 |
| ETTm2 | 96 | **0.155** | **0.240** | 0.161 | 0.248 | 0.163 | 0.251 | 0.175 | 0.266 | 0.164 | 0.250 | 0.157 | 0.247 | 0.165 | 0.256 | 0.165 | 0.255 | 0.296 | 0.391 | 0.164 | 0.255 |
| | 192 | **0.211** | **0.283** | 0.214 | 0.287 | 0.219 | 0.290 | 0.242 | 0.312 | 0.219 | 0.288 | 0.214 | 0.286 | 0.225 | 0.298 | 0.221 | 0.293 | 0.369 | 0.416 | 0.224 | 0.304 |
| | 336 | **0.258** | **0.314** | 0.267 | 0.320 | 0.269 | 0.330 | 0.282 | 0.337 | 0.267 | 0.319 | 0.269 | 0.322 | 0.277 | 0.332 | 0.276 | 0.327 | 0.588 | 0.600 | 0.277 | 0.337 |
| | 720 | **0.342** | **0.368** | 0.348 | 0.374 | 0.349 | 0.382 | 0.375 | 0.394 | 0.361 | 0.377 | 0.363 | 0.382 | 0.360 | 0.387 | 0.362 | 0.381 | 0.750 | 0.612 | 0.371 | 0.401 |
| Weather | 96 | **0.141** | **0.186** | 0.146 | 0.191 | 0.147 | 0.196 | 0.157 | 0.207 | 0.148 | 0.195 | 0.166 | 0.222 | 0.147 | 0.198 | 0.150 | 0.200 | 0.143 | 0.210 | 0.170 | 0.230 |
| | 192 | **0.186** | **0.230** | 0.188 | 0.231 | 0.193 | 0.240 | 0.200 | 0.248 | 0.191 | 0.235 | 0.213 | 0.259 | 0.192 | 0.243 | 0.191 | 0.239 | 0.195 | 0.261 | 0.216 | 0.273 |
| | 336 | **0.233** | **0.272** | 0.234 | 0.268 | 0.245 | 0.280 | 0.252 | 0.287 | 0.243 | 0.274 | 0.262 | 0.291 | 0.247 | 0.284 | 0.242 | 0.279 | 0.254 | 0.319 | 0.258 | 0.307 |
| | 720 | 0.312 | 0.331 | 0.305 | 0.319 | 0.323 | 0.334 | 0.320 | 0.336 | 0.318 | 0.326 | 0.329 | 0.338 | 0.318 | 0.330 | 0.312 | 0.330 | 0.335 | 0.385 | 0.323 | 0.362 |
| Solar | 96 | **0.169** | 0.214 | **0.169** | **0.195** | 0.181 | 0.247 | 0.190 | 0.244 | 0.218 | 0.235 | 0.201 | 0.252 | 0.179 | 0.232 | 0.170 | 0.234 | 0.183 | 0.208 | 0.199 | 0.265 |
| | 192 | **0.177** | 0.231 | 0.187 | **0.207** | 0.199 | 0.257 | 0.193 | 0.257 | 0.196 | 0.220 | 0.221 | 0.261 | 0.201 | 0.259 | 0.204 | 0.302 | 0.208 | 0.226 | 0.220 | 0.282 |
| | 336 | **0.186** | 0.238 | 0.199 | **0.213** | 0.208 | 0.269 | 0.203 | 0.266 | 0.195 | 0.220 | 0.233 | 0.269 | 0.190 | 0.256 | 0.212 | 0.293 | 0.212 | 0.239 | 0.234 | 0.295 |
| | 720 | **0.193** | 0.248 | 0.202 | **0.216** | 0.212 | 0.275 | 0.223 | 0.281 | 0.208 | 0.237 | 0.236 | 0.271 | 0.203 | 0.261 | 0.215 | 0.307 | 0.215 | 0.256 | 0.243 | 0.301 |
| Electricity | 96 | **0.122** | **0.215** | 0.128 | 0.219 | 0.128 | 0.222 | 0.134 | 0.230 | 0.135 | 0.222 | 0.126 | 0.221 | 0.153 | 0.256 | 0.143 | 0.247 | 0.134 | 0.231 | 0.140 | 0.237 |
| | 192 | **0.140** | **0.234** | 0.145 | 0.235 | 0.147 | 0.242 | 0.154 | 0.250 | 0.157 | 0.253 | 0.144 | 0.239 | 0.168 | 0.269 | 0.158 | 0.260 | 0.146 | 0.243 | 0.154 | 0.251 |
| | 336 | **0.157** | **0.252** | 0.163 | 0.255 | 0.165 | 0.260 | 0.169 | 0.265 | 0.170 | 0.267 | 0.161 | 0.253 | 0.189 | 0.291 | 0.168 | 0.267 | 0.165 | 0.264 | 0.169 | 0.268 |
| | 720 | **0.191** | **0.284** | 0.193 | 0.281 | 0.199 | 0.289 | 0.194 | 0.288 | 0.211 | 0.302 | 0.199 | 0.286 | 0.228 | 0.320 | 0.214 | 0.307 | 0.237 | 0.314 | 0.204 | 0.301 |
| Traffic | 96 | 0.362 | 0.247 | **0.360** | **0.238** | 0.368 | 0.252 | 0.363 | 0.265 | 0.384 | 0.250 | 0.389 | 0.276 | 0.369 | 0.257 | 0.370 | 0.262 | 0.526 | 0.288 | 0.395 | 0.275 |
| | 192 | **0.379** | 0.254 | 0.383 | **0.249** | 0.382 | 0.261 | 0.384 | 0.273 | 0.405 | 0.257 | 0.406 | 0.280 | 0.400 | 0.272 | 0.386 | 0.269 | 0.503 | 0.263 | 0.407 | 0.280 |
| | 336 | **0.390** | 0.258 | 0.395 | 0.259 | 0.393 | 0.268 | 0.396 | 0.277 | 0.424 | 0.265 | 0.425 | 0.291 | 0.407 | 0.272 | 0.396 | 0.275 | 0.505 | 0.276 | 0.417 | 0.286 |
| | 720 | 0.424 | 0.281 | 0.435 | **0.278** | 0.438 | 0.297 | 0.445 | 0.308 | 0.452 | 0.283 | 0.450 | 0.303 | 0.461 | 0.316 | 0.435 | 0.295 | 0.552 | 0.301 | 0.454 | 0.308 |

**Baselines.** We compare MoFo with 17 advanced baselines in long-term time series forecasting comprising DUET [37], PDF [7], iTransformer [24], Pathformer [5], CycleNet [20], TimeMixer [59], PatchTST [32], Crossformer [75], DLinear [73], NLinear [73], FITS [68], FiLM [78], MICN [57], FEDformer [79], Triformer [6], Non-stationary Transformer [23], and Informer [77].

## 4.2 Forecasting Performance of MoFo

Experimental results are summarized in Table 2. Due to space constraints, we compare against a larger set of baselines in Appendix D.3. CycleNet explicitly models periodic patterns, yet achieves lower forecasting performance compared to DLinear, which merely employs simple fully connected layers. TimeMixer, on the other hand, utilizes multi-scale modeling techniques to comprehensively capture complex temporal dynamics. Compared to methods such as PatchTST, Pathformer, and other approaches that rely on continuous patching strategies, our model achieves superior forecasting performance. This improvement is attributed to the proposed Periodicity-based Discrete Patching, which enables the model to better capture temporal dependencies across time steps. PDF introduces a multi-scale decomposition framework that models temporal dependencies from both long-term and short-term perspectives. DUET employs bidirectional clustering over both temporal and channel dimensions to adaptively capture spatio-temporal dependencies, achieving the best performance among existing baselines. However, it does not explicitly model periodic patterns and suffers from high computational complexity. Benefiting from the Period-Aware Modulator, our model focuses explicitly on periodic pattern learning, leading to the overall best forecasting accuracy. These results demonstrate the effectiveness of our design choices in capturing long-range periodic dependencies.

## 4.3 Efficiency Analysis of MoFo

We compare the computational efficiency with Transformer-based baselines on the Traffic dataset. As shown in Table 3 and Table 11, MoFo achieves the best forecasting accuracy among all Transformer-based models, while demonstrating the lowest computational complexity and highest efficiency. Similarly, PatchTST, which also adopts a patching strategy, employs an independent channel learning approach that significantly increases model complexity. Its parameter count grows by more than $10\times$, FLOPs increase by $25\times$, and training speed slows down by $17\times$. Pathformer enhances prediction accuracy through dynamic path adaptation and a patching strategy, albeit at the cost of increased learning overhead. Compared to DUET, one of the top-performing baseline models, MoFo reduces the number of parameters by more than $3\times$, the computational cost by more than $10\times$, and significantly lowers both memory consumption and training time. This is because our model's complexity grows quadratically with the period length, which is significantly shorter than the input sequence length. Moreover, we show that only a single Transformer layer is sufficient for effective learning.

Table 3: Efficiency comparison of MoFo and SOTA baselines with $L = 720$ in Traffic dataset. All results of each model are under the optimal hyperparameters for fair comparison. Parameters: All learnable parameters requiring gradient descent. MACs: multiply–accumulate operations. FLOPs: floating point operations. M: Million ($10^6$). B: Billion ($10^9$). T: Trillion ($10^{12}$). MB: Megabyte. s: Second. ↑ indicates the relative percentage increasing regarding MoFo.

| | Models | MSE | # Parameters | # MACs | # FLOPs | Memory Usage | Epoch Time |
|---|---|---|---|---|---|---|---|
| Traffic [L = 720] | Crossformer | $0.552_{\uparrow30.18\%}$ | $3.23\,M_{\uparrow34.58\%}$ | $85.03\,B_{\uparrow13.43\%}$ | $92.41\,B_{\uparrow20.94\%}$ | $13,556\,MB_{\uparrow171.77\%}$ | $368\,s_{\uparrow682.97\%}$ |
| | PatchTST | $0.435_{\uparrow2.59\%}$ | $27.8\,M_{\uparrow1058.75\%}$ | $2.02\,T_{\uparrow2594.77\%}$ | $2.08\,T_{\uparrow2622.16\%}$ | $44,782\,MB_{\uparrow797.79\%}$ | $876\,s_{\uparrow1763.83\%}$ |
| | Pathformer | $0.452_{\uparrow6.60\%}$ | $9.61\,M_{\uparrow300.42\%}$ | $110.46\,B_{\uparrow47.36\%}$ | $117.76\,B_{\uparrow54.12\%}$ | $36,602\,MB_{\uparrow633.80\%}$ | $1,081\,s_{\uparrow2200.0\%}$ |
| | iTransformer | $0.445_{\uparrow4.95\%}$ | $5.37\,M_{\uparrow123.75\%}$ | $297.96\,B_{\uparrow297.49\%}$ | $446.30\,B_{\uparrow484.09\%}$ | $19,608\,MB_{\uparrow293.10\%}$ | $52\,s_{\uparrow10.64\%}$ |
| | PDF | $0.438_{\uparrow3.30\%}$ | $2.45\,M_{\uparrow2.08\%}$ | $637.05\,B_{\uparrow749.85\%}$ | $662.47\,B_{\uparrow766.99\%}$ | $38,014\,MB_{\uparrow662.11\%}$ | $76\,s_{\uparrow61.70\%}$ |
| | DUET | $0.435_{\uparrow2.59\%}$ | $11.2\,M_{\uparrow367.08\%}$ | $137.33\,B_{\uparrow83.20\%}$ | $975.96\,B_{\uparrow1177.27\%}$ | $75,616\,MB_{\uparrow1415.96\%}$ | $516\,s_{\uparrow997.87\%}$ |
| | **MoFo** | **0.424** | **2.40 M** | **74.96 B** | **76.74 B** | **4,988 MB** | **47 s** |

## 4.4 Hyperparameters Sensitivity Experiments

We investigate the sensitivity of MoFo to its two core hyperparameters—the number of Transformer layers and model dimensionality—on the ETTh2 and Electricity datasets. For each forecasting horizon $L \in \{96, 192, 336, 720\}$, we use the best-performing hyperparameter configuration and vary only the target hyperparameter, as shown in Figure 4.4. We report both MSE and MAE for evaluation. The number of Transformer layers is varied from 1 to 6, and the model dimensionality is tested in

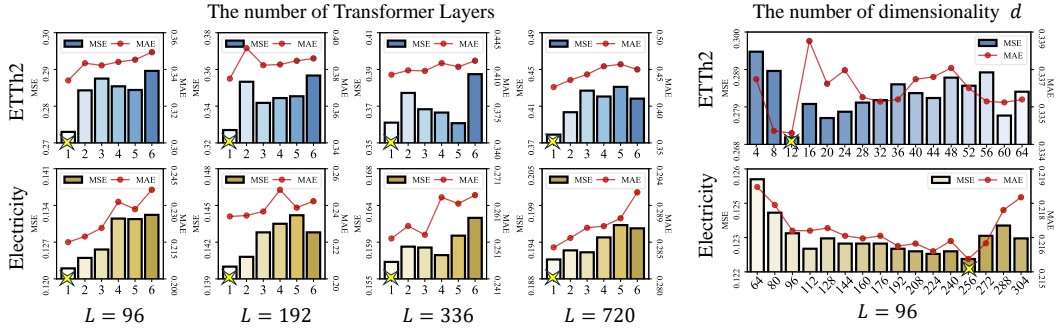

Figure 3: Hyperparameter sensitivity experiments.

Figure 4: Ablation study on all datasets with forecasting horizons $L = 712$.

the range of $4 \sim 64$ (ETTh2) and $64 \sim 304$ (Electricity), with the number of attention heads fixed at $H = 4$. Results show that MoFo achieves strong performance even with a single Transformer layer, while additional layers yield marginal gains. This suggests that the model efficiently captures inter-patch dependencies without requiring deep architectures. Furthermore, increasing the dimensionality generally improves performance; however, the gains plateau beyond a certain threshold, indicating diminishing returns from further capacity expansion.

## 4.5 Ablation Study

We design ablation experiments to validate the soundness of the each component of MoFo: '**+ Cpatch**' which used continuous patch technology; '**+Sinuous Pos**' and '**+ Learnable Pos**' which use Sinuous and Learnable position instead of our periodic relative position, respectively; '**- Modulator**' which removes the period-aware modulator; '**+ Mean Loss**' which only use L1 loss function. As shown in Figure 3, the ablation study reveals that '-Modulator' variant achieves the worst performance. This is because the Period-Aware Modulator plays a crucial role in guiding the model to focus on extracting periodic patterns. '+cpatch' variant also suffers from higher prediction errors, which can be attributed to the fact that our proposed discrete patching strategy enables direct modeling of dependencies among periodically aligned time steps. In summary, all variants perform worse than MoFo model to varying degrees, demonstrating the effectiveness and necessity of each component in capturing long-range periodic dependencies.

## 4.6 Scalability for Look-back Window of MoFo

We evaluate the scalability of MoFo under varying look-back window lengths. Specifically, we construct an ultra-long look-back setting on the ETTm2 dataset, where the input sequence length reaches up to 10K, one hundred times of the forecasting horizon: $L = 96, T = 96 * (5k), k = \{1, 2, \ldots, 20\}$. We report the maximum memory consumption during training, per-epoch training time, and FLOPs. As shown in Figure 5, as the look-back window increases, both DUET and PatchTST exhibit a sharp rise in memory usage and training time, indicating that their computational complexity scales closely with the input length. In contrast, even when the input sequence length grows by $100\times$, MoFo sees less than a $10\%$ increase in peak memory usage and less than a $2\%$ increase in training time. Its computational cost depends primarily on the period length rather than

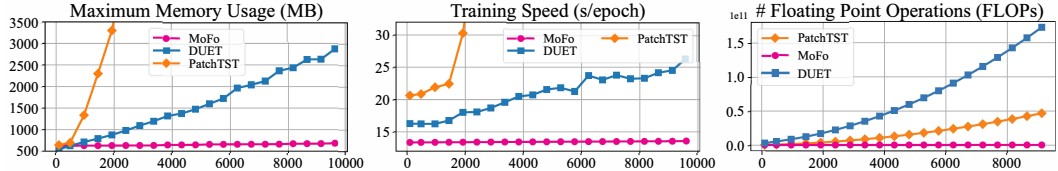

Figure 5: Scalability for look-back window of models on ETTm2 dataset.

the full sequence length. These results highlight the strong scalability of MoFo and demonstrate its great potential for modeling ultra-long sequences with minimal overhead.

### 4.7 Attention Visualization

We extract and visualize the attention coefficient matrices from models trained on four datasets with periodic lengths ($P$) of 24, 96, 144, and 24, respectively, all spanning a time window of one day. As shown in Figure 6, the coefficients in each attention head exhibit distinct periodic patterns. Notably, when the cyclical distance between two time steps is the largest (i.e., equal to $P/2$), the attention scores reach their minimum values. These attention coefficients that align with the underlying periodic trends significantly enhance the model's performance in long-term time series forecasting.



Figure 6: Visualization of attention scores.

## 5 Conclusion

In this paper, we propose MoFo, a novel framework for time series forecasting that leverages the inherent periodic structure of temporal data to explicitly model both periodic correlations and temporal trends. Through our proposed Period-structured patches, the model is able to directly capture correlations among time steps that share the same phase across periods. We further introduce a period-aware modulator, which enhances the attention mechanism with an adaptive inductive bias guided by underlying periodic trends. Experimental results demonstrate that MoFo achieves competitive forecasting performance compared to state-of-the-art methods, while significantly improving computational efficiency and scalability, especially for long input sequences.

## Acknowledgment

This paper is partially supported by the National Natural Science Foundation of China (No.12227901). The AI-driven experiments, simulations and model training were performed on the robotic AI-Scientist platform of Chinese Academy of Sciences.

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

# A  Mathematics Justification

## A.1  The Proof of Theorem 1

**Proof of (1):** For arbitrary $\alpha, \beta > 0$, there are $\mathcal{S}\left(0; \alpha, \beta\right) = 1, \mathcal{S}\left(+\infty; \alpha, \beta\right) = 0$.

In fact,

$$
\begin{aligned}
\mathcal{S}\left(0; \alpha, \beta\right) &= \frac{1}{1 + \exp\left(\alpha\left(0 - \beta\right)\right)} + \frac{\exp\left(-0\right)}{1 + \exp\left(\alpha\beta\right)} = \frac{1}{1 + \exp\left(-\alpha\beta\right)} + \frac{1}{1 + \exp\left(\alpha\beta\right)} \\
&= \frac{\exp\left(\alpha\beta\right)}{1 + \exp\left(\alpha\beta\right)} + \frac{1}{1 + \exp\left(\alpha\beta\right)} = 1.
\end{aligned}
\tag{16}
$$

and

$$
\mathcal{S}\left(+\infty; \alpha, \beta\right) = \frac{1}{1 + \exp\left(\alpha\left(+\infty - \beta\right)\right)} + \frac{\exp\left(-\infty\right)}{1 + \exp\left(\alpha\beta\right)} = \frac{1}{1 + \infty} + \frac{0}{1 + \exp\left(\alpha\beta\right)} = 0. \tag{17}
$$

$\square$

**Proof of (2):** The upper error bound of the smooth approximation satisfies,

$$
\int_0^{+\infty} \left|\mathcal{H}\left(\beta - \gamma\right) - \mathcal{S}\left(\gamma; \alpha, \beta\right)\right| d\gamma \leq \frac{\log 2}{\alpha} - \frac{1}{1 + \exp\left(\alpha\right)} \to 0 \quad (\alpha \to +\infty). \tag{18}
$$

In fact, the Heaviside step function $\mathcal{H}(\beta - \gamma)$ can be viewed as the differentiation of the maximum value function $\max\{0, \beta - \gamma\}$ as follows,

$$
\mathcal{H}(\beta - \gamma) = \frac{d}{d(\beta - \gamma)} \max\{0, \beta - \gamma\} = -\frac{d}{d\gamma} \max\{0, \beta - \gamma\}. \tag{19}
$$

Our sigmoidal attenuation function can be seen as follows,

$$
\mathcal{S}\left(\gamma; \alpha, \beta\right) = -\frac{d}{d\gamma} \frac{1}{\alpha} \log\left(1 + \exp\left(\alpha(\beta - \gamma)\right)\right) - \frac{d}{d\gamma} \frac{\exp\left(-\gamma\right)}{1 + \exp\left(\alpha\beta\right)}. \tag{20}
$$

Hence, the cumulative error $\int_0^{+\infty} \left|\mathcal{H}\left(\beta - \gamma\right) - \mathcal{S}\left(\gamma; \alpha, \beta\right)\right| d\gamma$ satisfies,

$$
\begin{aligned}
&\int_0^{+\infty} \left|\mathcal{H}\left(\beta - \gamma\right) - \mathcal{S}\left(\gamma; \alpha, \beta\right)\right| d\gamma \\
&= \int_0^{\beta} \left|\mathcal{H}\left(\beta - \gamma\right) - \mathcal{S}\left(\gamma; \alpha, \beta\right)\right| d\gamma + \int_{\beta}^{+\infty} \left|\mathcal{H}\left(\beta - \gamma\right) - \mathcal{S}\left(\gamma; \alpha, \beta\right)\right| d\gamma \\
&= \int_0^{\beta} \mathcal{H}\left(\beta - \gamma\right) - \mathcal{S}\left(\gamma; \alpha, \beta\right) d\gamma + \int_{\beta}^{+\infty} -\mathcal{H}\left(\beta - \gamma\right) + \mathcal{S}\left(\gamma; \alpha, \beta\right) d\gamma \\
&= \int_0^{\beta} -\frac{d}{d\gamma} \max\{0, \beta - \gamma\} + \frac{d}{d\gamma} \frac{1}{\alpha} \log\left(1 + \exp\left(\alpha(\beta - \gamma)\right)\right) + \frac{d}{d\gamma} \frac{\exp\left(-\gamma\right)}{1 + \exp\left(\alpha\beta\right)} d\gamma \\
&\quad + \int_{\beta}^{+\infty} \frac{d}{d\gamma} \max\{0, \beta - \gamma\} - \frac{d}{d\gamma} \frac{1}{\alpha} \log\left(1 + \exp\left(\alpha(\beta - \gamma)\right)\right) - \frac{d}{d\gamma} \frac{\exp\left(-\gamma\right)}{1 + \exp\left(\alpha\beta\right)} d\gamma \\
&= \left(-\max\{0, \beta - \gamma\} + \frac{1}{\alpha} \log\left(1 + \exp\left(\alpha(\beta - \gamma)\right)\right) + \frac{\exp\left(-\gamma\right)}{1 + \exp\left(\alpha\beta\right)}\right)\Bigg|_0^{\beta} \\
&\quad + \left(\max\{0, \beta - \gamma\} - \frac{1}{\alpha} \log\left(1 + \exp\left(\alpha(\beta - \gamma)\right)\right) - \frac{\exp\left(-\gamma\right)}{1 + \exp\left(\alpha\beta\right)}\right)\Bigg|_{\beta}^{+\infty} \\
&= \left(-0 + \frac{\log 2}{\alpha} + \frac{\exp\left(-\beta\right)}{1 + \exp\left(\alpha\beta\right)} + \beta - \frac{1}{\alpha} \log\left(1 + \exp\left(\alpha\beta\right)\right) - \frac{1}{1 + \exp\left(\alpha\beta\right)}\right) \\
&\quad + \left(0 - 0 - 0 - 0 + \frac{\log 2}{\alpha} + \frac{\exp\left(-\beta\right)}{1 + \exp\left(\alpha\beta\right)}\right) \\
&= \frac{2\log 2}{\alpha} + \frac{2\exp\left(-\beta\right) - 1}{1 + \exp\left(\alpha\beta\right)} + \beta - \frac{1}{\alpha} \log\left(1 + \exp\left(\alpha\beta\right)\right).
\end{aligned}
\tag{21}
$$

Since,

$$\beta - \frac{1}{\alpha} \log \left(1 + \exp \left(\alpha\beta\right)\right) < \beta - \frac{1}{\alpha} \log \left(\exp \left(\alpha\beta\right)\right) = \beta - \frac{\alpha\beta}{\alpha} = \beta - \beta = 0, \qquad (22)$$

and

$$\frac{2\log 2}{\alpha} + \frac{2\exp\left(-\beta\right) - 1}{1 + \exp\left(\alpha\beta\right)} \leq \frac{2\log 2}{\alpha} + \frac{1}{1 + \exp\left(\alpha\beta\right)} \leq \frac{2\log 2}{\alpha} + \frac{1}{1 + \exp\left(\alpha\right)}, \qquad (23)$$

then the the cumulative error has upper bound satisfies

$$\int_0^{+\infty} |\mathcal{H}\left(\beta - \gamma\right) - \mathcal{S}\left(\gamma; \alpha, \beta\right)| \, \mathrm{d}\gamma < \frac{2\log 2}{\alpha} + \frac{1}{1 + \exp\left(\alpha\right)} \to 0^+ \quad (\alpha \to +\infty). \qquad (24)$$

$\square$

## A.2  Autocorrelation Function and Bartlett's Test with Null Hypothesis

**Autocorrelation Function (ACF).** The Autocorrelation Function [3] measures the linear correlation between a time series and a lagged version of itself. For the complete observable time series values $\mathbf{X} \in \mathbb{R}^{N_t}$ with total observable time steps $N_t$, the autocorrelation function $\mathcal{A}_k$ of $\mathbf{X}$ at lag $k \geq 0$ is the ratio of the estimator of the covariance between the time series and the series lagged by k to the estimator of the variance of the time series as follows:

$$\mathcal{A}_k = \frac{\sum_{t=1}^{N_t - k}(\mathbf{X}_t - \mu)(\mathbf{X}_{t+k} - \mu)}{\sum_{t=1}^{N_t}(\mathbf{X}_t - \mu)^2} \in [-1, 1], \qquad (25)$$

where $\mu = \frac{1}{N_t}\sum_{t=1}^{N_t} \mathbf{X}_t \in \mathbb{R}$ is the expected value of $\mathbf{X}$.

**Bartlett's Test with Null Hypothesis**. The autocorrelation coefficients $\mathcal{A}_k$ can be viewed as random variables. However, even in the time series consisting of pure random noise, there may exist non-zero autocorrelation coefficients at some lags [39]. Hence, we require a method to ascertain whether an observed autocorrelation $\mathcal{A}_k$ represents a truly non-zero population or is merely due to this inherent randomness. This is typically achieved by performing confidence intervals derived from Bartlett's test [2] on a null hypothesis [38]. The null hypothesis denoted as $\mathcal{H}_0$ is a fundamental concept in statistical inference [1]. Its primary purpose is to serve as a base assumption for hypothesis testing. In the context of ACF, the relevant null hypothesis is $\mathcal{H}_0 : \mathcal{A}_k = 0$ that the ground-true autocorrelation coefficient $\mathcal{A}_k$ at a specific lag $k$ is zero. The Bartlett's test constructs confidence intervals to test this null hypothesis for individual lags. These intervals are based on an estimate of the standard deviation of $\mathcal{A}_k$ under the assumption that $\mathcal{H}_0$ is true. Specifically, the $1 - \alpha$ confidence interval[2] for the autocorrelation $\mathcal{A}_k$ under the null hypothesis $\mathcal{H}_0 : \mathcal{A}_k = 0$ is centered at 0, with boundaries given by $\pm Z_{\alpha/2} \cdot \mathrm{SE}(\mathcal{A}_k)$. Using Bartlett's formula for the variance of $\mathcal{A}_k$, the standard error $\mathrm{SE}(\mathcal{A}_k)$ is approximated by:

$$\mathrm{SE}(\mathcal{A}_k) \approx \sqrt{\frac{1}{N_t}(1 + 2\sum_{j=1}^{k-1} \mathcal{A}_j^2)}, \qquad (26)$$

where $\mathcal{A}_j$ are the autocorrelations for lags $j = 1, 2, \ldots, k - 1$ and $Z_{\alpha/2}$ is the $1 - \alpha/2$ quantile of the standard normal distribution (e.g., $Z_{0.025} \approx 1.96$ for a 95% confidence interval, corresponding to $\alpha = 0.05$) [61]. Thus, the approximate $1 - \alpha$ confidence interval boundaries for testing $H_0 : \mathcal{A}_k = 0$ are:

$$\left[ -Z_{\alpha/2}\sqrt{\frac{1}{N_t}(1 + 2\sum_{j=1}^{k-1} \mathcal{A}_j^2)}, +Z_{\alpha/2}\sqrt{\frac{1}{N_t}(1 + 2\sum_{j=1}^{k-1} \mathcal{A}_j^2)} \right]. \qquad (27)$$

Any autocorrelation value $\mathcal{A}_k$ that falls within this confidence interval is considered consistent with the null hypothesis ($\mathcal{H}_0 : \mathcal{A}_k = 0$). In such cases, we do not have sufficient statistical evidence to

---

[2]It is important to note that the alpha $\alpha$ here is different from the learnable parameter alpha mentioned in the RRF in the MoFo. The alpha $\alpha$ here is a specific parameter notation used in statistics [41]. We maintain consistency here to reduce potential confusion with specialized terminology.

conclude that the true autocorrelation at lag $k$ is different from zero; the observed $\mathcal{A}_k$ is likely due to random variation inherent in a noise process. Conversely, if $\mathcal{A}_k$ exceeds these boundaries, we reject the null hypothesis and conclude that the autocorrelation at lag $k$ is statistically significant.

We present in the Fig 7 the average ACF values and from some selective channel for all datasets used in this study, calculated across each channel with a maximum lag of 512, and report the null hypothesis region with $95\%$ confidence interval. We observe that at the same positions within each period of the time series, the ACF values exhibit significant peaks. Therefore, it is highly rational to apply patching based on periodic positions for the time steps. The orange region represents the 95% confidence interval for the null hypothesis from Bartlett's Test [2]. Correlation coefficients that fall within this interval are not statistically significant and cannot be rejected as noise [4]

## B    Related Work

Time series modeling serves as a core task in numerous domains, such as transportation and atmospheric science [21, 29, 44, 45, 46, 47, 48, 49, 69]. Early approaches primarily relied on recurrent neural networks (RNNs) and temporal convolutional networks (TCNs). In recent years, models based on multilayer perceptrons (MLPs) have garnered significant attention due to their lightweight and efficient performance.

**RNN-based Models.** RNNs, among the earliest deep learning architectures for sequential data, have been widely adopted for long-term time series forecasting, with notable variants such as LSTM [76] and GRU [8]. To mitigate the problem of too many recurrent steps, SegRNN [19] introduces a segmented recurrence mechanism combined with parallel multi-step prediction, substantially cutting down the number of iterations.

**TCN-Based Model.** TCNs employ convolutional operations to effectively model local contextual patterns in time series, offering a good trade-off between computational efficiency and forecasting accuracy. Recent advances have extended TCNs to better capture long-range temporal dependencies. For instance, ModernTCN [27] adopts large convolutional kernels to greatly expand the receptive field, allowing the model to capture broader temporal structures. Likewise, Pyraformer [22] integrates TCN layers with a Transformer framework; it uses stacked TCN layers for downsampling to obtain coarse-grained time series representations, which are then processed by the Transformer to enhance both scalability and performance.

**MLP-Based Model.** MLP-based models, when thoughtfully designed, have shown strong performance in time series forecasting. DLinear [73] illustrates this by using a moving average kernel to decompose the input series into trend and seasonal parts, each modeled separately by dedicated linear layers. PatchMLP [42] adopts a patching strategy that incorporates channel mixing to improve cross-variable information exchange. Extending this idea, HDMixer [11] employs adaptive patch lengths to capture both intra-patch short-term dynamics and inter-patch long-term dependencies while modeling intricate variable interactions. Meanwhile, FITS [68] operates MLPs in the frequency domain, leveraging spectral analysis to emphasize dominant signal components and better capture global temporal relationships.

## C    The Transformer Layer in MoFo

We use the pre-norm Transformer Layer [43, 56] of multi-head attention with Regulated Relaxation Function (Eq. 12).

**Vanilla Transformer Layer.** Transformer [43] consists of the self-attention function $\mathrm{MultiHead}(\cdot)$ with feedforward networks $\mathrm{FFN}(\cdot)$, and two distinct normalization layers $\mathrm{Norm}_i(\cdot)$. Assuming the input hidden representation is $\mathbf{Z} \in \mathbb{R}^{P \times d}$ with period length $P$ and model dimensionality $d$, the output hidden representation $\vec{\mathbf{Z}} \in \mathbb{R}^{n \times d}$ of one Transformer layer is as follows,

$$
\begin{aligned}
\vec{\mathbf{Z}} &= \mathrm{Multi\text{-}Head}(\mathrm{Norm}_2(\bar{\mathbf{Z}})) + \bar{\mathbf{Z}}, \\
\bar{\mathbf{Z}} &= \mathrm{FFN}(\mathrm{Norm}_1(\mathbf{Z})) + \mathbf{Z}.
\end{aligned}
\tag{28}
$$

Here we depict the pre-norm structure [56]. The multi-head attention mechanism is used in Transformer to improve the representation performance. Let $\tilde{\mathbf{Z}} = \mathrm{Norm}(\bar{\mathbf{Z}})$, the multi-head attention

function is a weighted combination of outputs from different head as follows,

$$\text{Multi-Head}(\tilde{\mathbf{Z}}) = \text{Concat}(\text{head}_1, \text{head}_2, \ldots, \text{head}_H)\mathbf{W}_O \in \mathbb{R}^{P \times d},$$

$$\text{head}_j = \text{Attention}(\tilde{\mathbf{Z}}\mathbf{W}_Q^j, \tilde{\mathbf{Z}}\mathbf{W}_K^j, \tilde{\mathbf{Z}}\mathbf{W}_V^j) \in \mathbb{R}^{P \times d_h}, \tag{29}$$

where $H$ is the number of heads. $\mathbf{W}_Q^j, \mathbf{W}_K^j, \mathbf{W}_V^j \in \mathbb{R}^{d \times d_h}$ and $\mathbf{W}_O \in \mathbb{R}^{d \times d}$ are learnable projections parameters with head dimensionality $d_h = d/H$. And self-attention function in vanilla Transformer is defined as follows,

$$\text{Attention}(\mathbf{Q}, \mathbf{K}, \mathbf{V}) = \text{Softmax}(\frac{\mathbf{Q}\mathbf{K}^\top}{\sqrt{d_h}})\mathbf{V} \in \mathbb{R}^{P \times P}, \tag{30}$$

where $\text{Softmax}$ is an exponential activation with $l_1$ normalization [9] in the last dimensionality. And the attention scores between the $i$-th token and all other tokens after softmax operation are as follows,

$$\text{Softmax}(\frac{\mathbf{Q}_i\mathbf{K}^\top}{\sqrt{d_h}}) = \frac{\exp\left(\frac{\mathbf{Q}_i\mathbf{K}^\top}{\sqrt{d_h}}\right)}{\sum_{j=1}^n \exp\left(\frac{\mathbf{Q}_i\mathbf{K}_j^\top}{\sqrt{d_h}}\right)} \in \mathbb{R}^P. \tag{31}$$

**Modification in MoFo.** However, we leverage Regulated Relaxation Function to instead the attention function in MoFo introduced in Section 3.2.2 as follows,

$$\text{Attention}(\mathbf{Q}, \mathbf{K}, \mathbf{V}) = \text{RRF}(\mathbf{Q}, \mathbf{K}, \mathbf{V}) = \text{Softmax}(\frac{\mathbf{Q}\mathbf{K}^\top}{\sqrt{d_h}} + \log \mathcal{S}(\mathbf{\Gamma}; \alpha, \beta))\mathbf{V} \in \mathbb{R}^{P \times d_h},$$

$$\text{where} \quad \mathbf{Q} = \mathbf{Z}\mathbf{W}_Q^j \in \mathbb{R}^{P \times d_h}, \mathbf{K} = \mathbf{Z}\mathbf{W}_K^j \in \mathbb{R}^{P \times d_h}, \mathbf{V} = \mathbf{Z}\mathbf{W}_V^j \in \mathbb{R}^{P \times d_h}, \tag{32}$$

The feedforward networks in MoFo are gated linear units [40] with Swish activation. Let $\dot{\mathbf{Z}} = \text{Norm}_1(\mathbf{Z})$, the $\text{FFN}(\cdot)$ is defined as follows,

$$\text{FFN}(\dot{\mathbf{Z}}) = \left(\text{SwiGLU}(\dot{\mathbf{Z}}\mathbf{W}_1 + \mathbf{b}_1) \odot (\dot{\mathbf{Z}}\mathbf{W}_2 + \mathbf{b}_2)\right)\mathbf{W}_3 + \mathbf{b}_3 \in \mathbb{R}^{P \times d}, \tag{33}$$

where weight matrices $\mathbf{W}_1, \mathbf{W}_2 \in \mathbb{R}^{d \times 4d}, \mathbf{W}_3 \in \mathbb{R}^{4d \times d}$ and bias parameters $\mathbf{W}_1, \mathbf{W}_2 \in \mathbb{R}^{4d}, \mathbf{W}_3 \in \mathbb{R}^d$ are learnable. The normalization layer is root mean square normalization [74] as follows,

$$\text{Norm}_1(\mathbf{Z}) = \frac{\mathbf{Z} \odot \mathbf{g}_1}{\sqrt{\frac{1}{d}\sum_{k=1}^d \mathbf{Z}_{:,k}}} \in \mathbb{R}^{P \times d}, \quad \text{Norm}_2(\bar{\mathbf{Z}}) = \frac{\bar{\mathbf{Z}} \odot \mathbf{g}_2}{\sqrt{\frac{1}{d}\sum_{k=1}^d \bar{\mathbf{Z}}_{:,k}}} \in \mathbb{R}^{P \times d}, \tag{34}$$

where $\mathbf{g}_1, \mathbf{g}_2 \in \mathbb{R}^d$ are learnable scale parameters of normalization.

# D Experiments

## D.1 Dataset Analysis

We further visualize the temporal correlations across multiple datasets. As shown in Figure 7, most datasets exhibit clear periodic patterns, with relatively high correlations between time steps separated by fixed intervals. This suggests that commonly adopted sequential input strategies—such as those used in patch-based methods—tend to group temporally adjacent but semantically unrelated time steps, thereby hindering the model's ability to capture intrinsic periodic structures.

## D.2 Settings

Our experiment is based on the TFB platform [34] for a fair comparison. Following the settings in TFB [34], we report the best performance within the optional historical sequence length $T \in \{96, 336, 512\}$ of the multiple forecasting length $L \in \{96, 192, 336, 720\}$ with two common generic metrics including the mean square error (MSE) and mean absolute error (MAE) to judge the performance of our model. Our experiments are executed on an NVIDIA A100 with 40GB memory. Our code environment is based on the PyTorch framework using Python 3.11.5. The 'Drop Last' trick is closed to ensure a fair comparison [18]. We adopt Adam [14] optimizer. The training process is guided by the $L_1$ loss function of the FreDF strategy [53]. The penalization distance parameter $\beta > 0$ is restricted in $(0, P)$. We utilize only one layer of Transformer with attention head $H = 4$ for each setting in all datasets. LTSF datasets often have multiple channels (or variates), and we adopt the channel independence approach [32] to simultaneous independent learning of all channels.

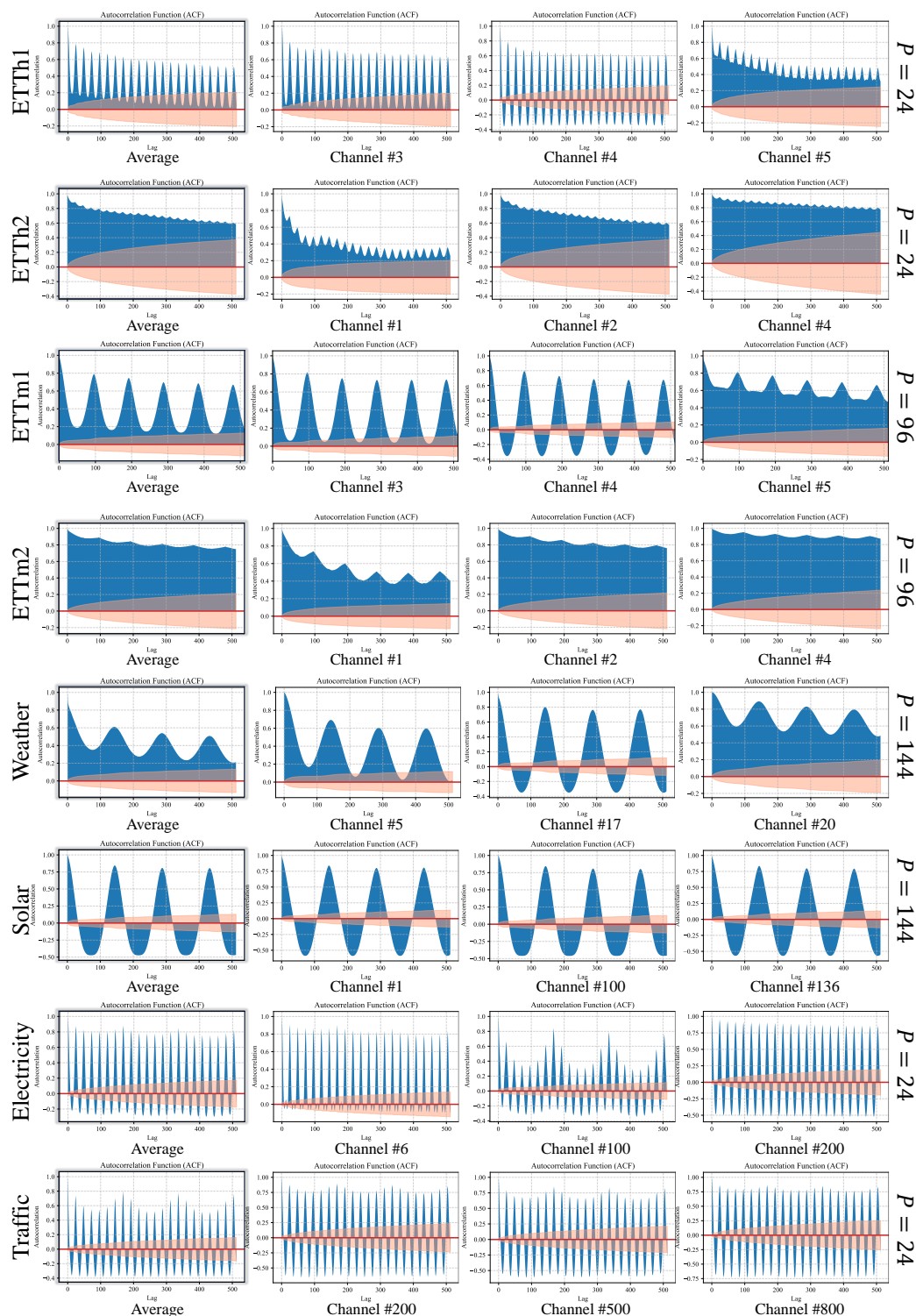

Figure 7: The autocorrelation visualization of all datasets.

Table 4: Performance comparisons for LTSF. The **best** results are marked in corresponding colors. All experimental results are selected from the best performance under the historical sequence length $T \in \{96, 336, 512\}$.

| Method | | MoFo (Ours) | | FITS (2023) | | Nlinear (2023) | | TimesNet (2023) | | FEDformer (2022) | | Triformer (2022) | | MICN (2022) | | FiLM (2022) | | Stationary (2022) | | Informer (2021) | |
|---|---|---|---|---|---|---|---|---|---|---|---|---|---|---|---|---|---|---|---|---|---|
| Metric | | MSE | MAE | MSE | MAE | MSE | MAE | MSE | MAE | MSE | MAE | MSE | MAE | MSE | MAE | MSE | MAE | MSE | MAE | MSE | MAE |
| ETTh1 | 96 | **0.360** | **0.389** | 0.376 | 0.396 | 0.385 | 0.403 | 0.389 | 0.412 | 0.379 | 0.419 | 0.399 | 0.425 | 0.378 | 0.412 | 0.370 | 0.394 | 0.591 | 0.524 | 0.571 | 0.399 |
| | 192 | **0.397** | **0.413** | 0.400 | 0.418 | 0.422 | 0.426 | 0.440 | 0.443 | 0.420 | 0.444 | 0.444 | 0.449 | 0.400 | 0.430 | 0.405 | 0.416 | 0.615 | 0.540 | 0.574 | 0.444 |
| | 336 | **0.407** | **0.424** | 0.419 | 0.435 | 0.431 | 0.429 | 0.523 | 0.487 | 0.458 | 0.466 | 0.492 | 0.479 | 0.428 | 0.447 | 0.434 | 0.435 | 0.632 | 0.551 | 0.588 | 0.492 |
| | 720 | 0.447 | 0.454 | **0.435** | 0.458 | 0.439 | **0.452** | 0.521 | 0.495 | 0.474 | 0.488 | 0.549 | 0.529 | 0.474 | 0.499 | 0.463 | 0.474 | 0.828 | 0.658 | 0.623 | 0.549 |
| ETTh2 | 96 | **0.273** | **0.334** | 0.277 | 0.345 | 0.276 | 0.338 | 0.334 | 0.370 | 0.337 | 0.380 | 0.936 | 0.660 | 0.313 | 0.372 | 0.282 | 0.346 | 0.347 | 0.387 | 0.394 | 0.936 |
| | 192 | **0.327** | **0.373** | 0.331 | 0.379 | 0.345 | 0.382 | 0.404 | 0.413 | 0.415 | 0.428 | 1.290 | 0.768 | 0.419 | 0.439 | 0.358 | 0.401 | 0.379 | 0.418 | 0.448 | 1.290 |
| | 336 | 0.361 | 0.405 | **0.350** | **0.396** | 0.368 | 0.408 | 0.389 | 0.435 | 0.389 | 0.457 | 1.325 | 0.781 | 0.474 | 0.475 | 0.372 | 0.425 | 0.358 | 0.413 | 0.464 | 1.325 |
| | 720 | **0.379** | **0.425** | 0.382 | 0.425 | 0.406 | 0.441 | 0.434 | 0.448 | 0.483 | 0.488 | 1.500 | 0.850 | 0.723 | 0.600 | 0.425 | 0.455 | 0.422 | 0.457 | 0.454 | 1.500 |
| ETTm1 | 96 | **0.286** | **0.335** | 0.303 | 0.345 | 0.301 | 0.343 | 0.340 | 0.378 | 0.463 | 0.463 | 0.349 | 0.388 | 0.303 | 0.349 | 0.301 | 0.343 | 0.415 | 0.410 | 0.422 | 0.349 |
| | 192 | **0.320** | **0.363** | 0.337 | 0.365 | 0.355 | 0.379 | 0.392 | 0.404 | 0.575 | 0.516 | 0.387 | 0.410 | 0.336 | 0.369 | 0.339 | 0.365 | 0.494 | 0.451 | 0.480 | 0.387 |
| | 336 | **0.347** | **0.382** | 0.368 | 0.384 | 0.372 | 0.385 | 0.423 | 0.426 | 0.618 | 0.544 | 0.426 | 0.446 | 0.370 | 0.391 | 0.374 | 0.385 | 0.577 | 0.490 | 0.531 | 0.426 |
| | 720 | **0.388** | **0.411** | 0.420 | 0.413 | 0.430 | 0.418 | 0.475 | 0.453 | 0.612 | 0.551 | 0.482 | 0.476 | 0.410 | 0.421 | 0.423 | 0.414 | 0.636 | 0.535 | 0.578 | 0.482 |
| ETTm2 | 96 | **0.155** | **0.240** | 0.165 | 0.254 | 0.163 | 0.252 | 0.189 | 0.265 | 0.216 | 0.309 | 0.276 | 0.344 | 0.173 | 0.271 | 0.165 | 0.254 | 0.210 | 0.294 | 0.302 | 0.276 |
| | 192 | **0.211** | **0.283** | 0.219 | 0.291 | 0.218 | 0.290 | 0.254 | 0.310 | 0.297 | 0.360 | 0.473 | 0.453 | 0.232 | 0.313 | 0.220 | 0.291 | 0.338 | 0.373 | 0.365 | 0.473 |
| | 336 | **0.258** | **0.314** | 0.272 | 0.326 | 0.273 | 0.326 | 0.313 | 0.345 | 0.366 | 0.400 | 0.692 | 0.549 | 0.303 | 0.367 | 0.277 | 0.329 | 0.432 | 0.416 | 0.414 | 0.692 |
| | 720 | **0.342** | **0.368** | 0.359 | 0.381 | 0.361 | 0.382 | 0.413 | 0.402 | 0.459 | 0.450 | 1.936 | 0.856 | 0.467 | 0.477 | 0.363 | 0.386 | 0.554 | 0.476 | 0.468 | 1.936 |
| Weather | 96 | **0.141** | **0.186** | 0.172 | 0.225 | 0.180 | 0.226 | 0.168 | 0.214 | 0.229 | 0.298 | 0.170 | 0.236 | 0.172 | 0.232 | 0.178 | 0.229 | 0.188 | 0.242 | 0.256 | 0.170 |
| | 192 | **0.186** | **0.230** | 0.215 | 0.261 | 0.218 | 0.261 | 0.219 | 0.262 | 0.265 | 0.334 | 0.216 | 0.277 | 0.214 | 0.271 | 0.218 | 0.263 | 0.240 | 0.290 | 0.300 | 0.216 |
| | 336 | **0.233** | **0.272** | 0.261 | 0.295 | 0.266 | 0.296 | 0.278 | 0.302 | 0.330 | 0.372 | 0.272 | 0.324 | 0.259 | 0.309 | 0.266 | 0.295 | 0.322 | 0.328 | 0.332 | 0.272 |
| | 720 | **0.312** | **0.331** | 0.326 | 0.341 | 0.334 | 0.345 | 0.353 | 0.351 | 0.423 | 0.418 | 0.350 | 0.379 | 0.309 | 0.343 | 0.332 | 0.341 | 0.396 | 0.378 | 0.388 | 0.350 |
| Solar | 96 | **0.169** | **0.214** | 0.208 | 0.255 | 0.202 | 0.245 | 0.198 | 0.270 | 0.485 | 0.570 | 0.225 | 0.279 | 0.190 | 0.25 | 0.214 | 0.259 | 0.365 | 0.390 | 0.368 | 0.225 |
| | 192 | **0.177** | **0.231** | 0.229 | 0.267 | 0.223 | 0.258 | 0.206 | 0.276 | 0.415 | 0.477 | 0.250 | 0.295 | 0.226 | 0.284 | 0.226 | 0.257 | 0.400 | 0.386 | 0.388 | 0.250 |
| | 336 | **0.186** | **0.238** | 0.241 | 0.273 | 0.238 | 0.265 | 0.208 | 0.284 | 1.008 | 0.839 | 0.261 | 0.297 | 0.259 | 0.308 | 0.241 | 0.265 | 0.414 | 0.394 | 0.420 | 0.261 |
| | 720 | **0.193** | **0.248** | 0.248 | 0.277 | 0.246 | 0.268 | 0.232 | 0.294 | 0.655 | 0.627 | 0.259 | 0.292 | 0.341 | 0.365 | 0.247 | 0.268 | 0.379 | 0.377 | 0.405 | 0.259 |
| Electricity | 96 | **0.122** | **0.215** | 0.139 | 0.237 | 0.140 | 0.236 | 0.169 | 0.271 | 0.191 | 0.305 | 0.201 | 0.298 | 0.158 | 0.266 | 0.154 | 0.246 | 0.171 | 0.274 | 0.321 | 0.201 |
| | 192 | **0.140** | **0.234** | 0.154 | 0.250 | 0.155 | 0.248 | 0.180 | 0.280 | 0.203 | 0.316 | 0.209 | 0.307 | 0.175 | 0.287 | 0.168 | 0.261 | 0.180 | 0.283 | 0.362 | 0.209 |
| | 336 | **0.157** | **0.252** | 0.170 | 0.268 | 0.171 | 0.264 | 0.204 | 0.304 | 0.221 | 0.333 | 0.225 | 0.323 | 0.184 | 0.296 | 0.189 | 0.284 | 0.204 | 0.305 | 0.416 | 0.225 |
| | 720 | **0.191** | **0.284** | 0.212 | 0.304 | 0.210 | 0.297 | 0.205 | 0.304 | 0.259 | 0.364 | 0.264 | 0.353 | 0.200 | 0.310 | 0.249 | 0.340 | 0.221 | 0.319 | 0.525 | 0.264 |
| Traffic | 96 | **0.362** | **0.247** | 0.407 | 0.290 | 0.395 | 0.272 | 0.595 | 0.312 | 0.593 | 0.365 | 0.589 | 0.323 | 0.517 | 0.313 | 0.412 | 0.284 | 0.603 | 0.330 | 0.392 | 0.589 |
| | 192 | **0.379** | **0.254** | 0.418 | 0.294 | 0.407 | 0.277 | 0.613 | 0.322 | 0.614 | 0.381 | 0.597 | 0.325 | 0.526 | 0.302 | 0.415 | 0.285 | 0.611 | 0.338 | 1.280 | 0.597 |
| | 336 | **0.390** | **0.258** | 0.433 | 0.308 | 0.417 | 0.282 | 0.626 | 0.332 | 0.627 | 0.389 | 0.617 | 0.332 | 0.545 | 0.307 | 0.430 | 0.299 | 0.628 | 0.342 | 0.477 | 0.617 |
| | 720 | **0.424** | **0.281** | 0.486 | 0.347 | 0.453 | 0.302 | 0.635 | 0.340 | 0.646 | 0.394 | 0.650 | 0.350 | 0.569 | 0.328 | 0.525 | 0.371 | 0.646 | 0.350 | 1.294 | 0.650 |

### D.3 Performance Comparison with More Baselines

Considering readability, we only compared our approach with some representative SOTA baselines in Section 4.2. Here, we include more additional LTSF baselines to provide a more comprehensive evaluation of the performance of MoFo. Specifically, we add the following baselines: MLP-based models including FITS [68] NLinear [73] and FiLM [78]; TCN-based Models including TimesNet [63] and MICN [57]; Transformer-based models including FEDformer [79], Triformer [6], Non-stationary Transformer (Stationary) [23] and Informer [77]. As shown in Table 4, NLinear remains a powerful baseline as a linear model. FITS and FiLM extract time series representation in the frequency domain and in combining Legendre memory models, respectively. MICN utilizes convolutional networks to capture local and global contexts, while FEDformer enhances the Transformer in the frequency domain. Triformer introduces efficient triangular attention with convolutional down-sampling on coarse-time series, and Non-stationary Transformer (Stationary) focuses on addressing the non-stationarity of time series. Finally, Informer effectively tackles the computational challenges of long sequence prediction through its ProbSparse attention and distillation techniques. However, MoFo demonstrates superiority across almost all metrics comparing to all the baselines by reasonable Period-based Discrete Patching strategy. The Modulator in MoFo not only dynamically models the periodicity of time series data but also addresses permutation invariance, enhancing the representation capabilities of the Transformer in MoFo.

### D.4 Performance on None Periodicity Datasets

To further evaluate the generalization ability of MoFo beyond strictly periodic signals, we investigate its performance on datasets that lack explicit periodicity of a publicly available Influenza-Like Illness (ILI) dataset released by the U.S. Centers for Disease Control and Prevention (CDC), which contains weekly reports of the proportion of ILI-related visits from 2002 to 2021. The ILI dataset exhibits weak or no clear seasonality, making it an appropriate dataset for testing models under

non-periodic temporal conditions. We forecast future prediction using four prediction horizons: $L \in \{24, 36, 48, 60\}$ with optional look-back window $T \in \{36, 104\}$.

To emphasize the modeling of non-periodic time series, we follow the preprocessing strategy of TimesNet [63] by identifying the main pseudo-period as the reciprocal of the dominant frequency derived via Fast Fourier Transform (FFT) for each input sequence. As shown in Table 5, although MoFo was originally designed to leverage explicit periodic structures, its superior results on ILI demonstrate strong adaptability and robustness in capturing complex temporal dependencies even in the absence of clear periodic patterns. This highlights that MoFo not only excels in periodic forecasting but also generalizes effectively to irregular, non-stationary time series domains.

Table 5: Performance comparison on non-periodicity dataset ILI.

| Methods | MoFo | | FITS | | TimesNet | |
|---|---|---|---|---|---|---|
| Metrics | MSE | MAE | MSE | MAE | MSE | MAE |
| 24 | **2.113** | **0.927** | 2.176 | 0.928 | 2.255 | 0.936 |
| 36 | **1.952** | **0.924** | 2.166 | 0.993 | 2.132 | 0.940 |
| 48 | **1.714** | **0.824** | 2.011 | 0.928 | 2.182 | 0.944 |
| 60 | **1.800** | **0.906** | 2.010 | 0.967 | 2.169 | 0.940 |

## D.5 Ablation Study on Padding Strategy

Our padding strategy, detailed in Section 3.1 formalized in Eq. 1, is designed to preserve temporal continuity when the sequence length $T$ is not an integer multiple of the detected period length $P$. Specifically, we start from the current time step and move backward to delineate periods of length $P$. Any prefix that does not form a complete period is left-padded with the leftmost elements of the nearest full period on its right. This scheme ensures that all time steps participate in subsequent computations without discarding boundary information. To evaluate the effect of this design, we introduce a variant termed **'+ Zero Padding'**, where incomplete periods are instead filled with zeros. We conduct a comparative analysis on the ETTm1 and ETTm2 datasets under identical settings. The results are summarized in Table 6, where the best-performing metrics are highlighted in bold.

Table 6: Ablation Study on Padding Strategy

| Methods | | MoFo | | + zeros padding | |
|---|---|---|---|---|---|
| Metric | | MSE | MAE | MSE | MAE |
| ETTm1 | 96 | **0.286** | **0.335** | 0.292 | 0.345 |
| | 192 | **0.320** | **0.363** | 0.328 | 0.372 |
| | 336 | **0.347** | **0.382** | 0.348 | 0.386 |
| | 720 | **0.388** | **0.411** | 0.401 | 0.435 |
| ETTm2 | 96 | **0.155** | **0.240** | 0.156 | 0.247 |
| | 192 | **0.211** | **0.283** | 0.215 | 0.293 |
| | 336 | **0.258** | **0.314** | 0.261 | 0.318 |
| | 720 | **0.342** | **0.368** | 0.345 | 0.372 |

Empirically, our proposed period-aware padding consistently outperforms zero padding across both benchmarks. We attribute this improvement to its ability to re-use the immediately preceding historical patterns, thereby maintaining local temporal coherence and facilitating smoother periodic transitions. In contrast, zero padding introduces abrupt discontinuities that disrupt the temporal rhythm, leading to inferior generalization. These findings confirm that our padding mechanism not only preserves data integrity but also serves as an implicit temporal regularizer that enhances long-term forecasting stability.

## D.6 Look-back Window Sensitivity Experiments

In time series forecasting, the look-back window length—i.e., the number of historical steps fed into the model—is a critical hyperparameter that directly affects performance. Different architectures exhibit varying sensitivities to the length of historical context: models emphasizing long-term dependencies may benefit from longer input sequences, while those designed for short-term dynamics

might suffer from redundant or noisy inputs when the window is excessively long. Therefore, treating the look-back length as a tunable hyperparameter rather than a fixed setting is essential for a fair and comprehensive evaluation.

We perform experiments under multiple look-back window configurations to systematically assess this sensitivity. In the original setup, look-back lengths are predefined for each dataset, and models are trained and tested under all candidate configurations. The best-performing results are then reported, reflecting each model's optimal temporal receptive field. This protocol ensures a fair comparison among models with heterogeneous design principles and differing dependency ranges.

In line with this methodology, we examine the performance of our model and baselines under varying look-back window lengths. For the forecasting horizon of 720, we adopt $\{96, 336, 512\}$ as candidate look-back windows, corresponding to commonly used temporal spans in long-term forecasting. The experimental results, summarized in Table 7, reveal that our model maintains consistently strong performance across different window lengths, demonstrating both its robustness and its capacity to adaptively leverage available historical information. These findings justify our choice of treating the look-back window length as a tunable hyperparameter in the main experiments and highlight the stability of MoFo under diverse temporal contexts.

Table 7: Look-back window sensitivity experiments of all optional look-back window length $T \in \{96, 336, 512\}$ on the forecasting length setting $L = 720$.

| Method | | MoFo | | FITS | | DLinear | | TimesNet | |
|---|---|---|---|---|---|---|---|---|---|
| Metric | | MSE | MAE | MSE | MAE | MSE | MAE | MSE | MAE |
| ETTh1 $T = 96$ | | **0.447** | **0.454** | 0.547 | 0.518 | 0.515 | 0.511 | 0.521 | 0.495 |
| ETTh1 $T = 336$ | | **0.459** | **0.469** | 0.475 | 0.487 | 0.471 | 0.493 | 0.542 | 0.519 |
| ETTh1 $T = 512$ | | 0.443 | 0.463 | **0.435** | **0.458** | 0.464 | 0.488 | 0.560 | 0.531 |
| ETTh2 $T = 96$ | | **0.416** | **0.442** | 0.439 | 0.452 | 0.650 | 0.571 | 0.434 | 0.448 |
| ETTh2 $T = 336$ | | **0.393** | **0.428** | 0.397 | 0.431 | 0.704 | 0.597 | 0.472 | 0.480 |
| ETTh2 $T = 512$ | | **0.379** | **0.425** | 0.382 | 0.425 | 0.786 | 0.623 | 0.480 | 0.468 |

## D.7 Period Sensitivity Experiments

The accurate identification of periodic structures plays a crucial role in MoFo's ability to capture long-term dependencies and recurrent temporal dynamics. However, in real-world time series, period selection is often uncertain due to noise, seasonal drift, or data heterogeneity. To comprehensively examine MoFo's robustness to such variations, we conduct three complementary sensitivity analyses: ❶ adjusting the given period length to test the impact of under- and over-estimation of period length, ❷ introducing multiple coexisting periodicities to evaluate the model's adaptability to mixed-period settings, and ❸ comparing robustness under complex dynamic period settings. The other experimental settings are kept consistent with the main experiments in Section 4.

❶ **Sensitivity to period length $P$.** To assess the importance of accurate period calibration in MoFo, we conduct a series of sensitivity experiments on the given period length $P$. We first design two variants—"+ Half" and "+ Double"—in which the detected period is halved or doubled, respectively. As summarized in Table 8, results on three representative datasets with distinct periodicities demonstrate that a well-calibrated period is crucial for forecasting accuracy. When $P$ is substantially under- or over-estimated, the model's ability to capture intrinsic temporal regularities degrades significantly, while the original configuration preserves the correct periodic structure and yields optimal performance. To further verify the robustness of this observation, we introduce four additional fine-grained perturbations: "+5%" and "+15%", where $P$ is increased by 5% and 15%, and "–5%" and "–15%", where $P$ is decreased by 5% and 15%. As shown in Table 8, even minor deviations from the detected period lead to measurable performance drops across datasets with different dominant periods. These results collectively highlight that the accuracy of period estimation plays a pivotal role in MoFo's ability to effectively model periodic dependencies in time series data.

❷ **Sensitivity to multiple coexisting periods.** In addition to single-period sensitivity, we further investigate MoFo's behavior under multiple periodic structures. We adopt the Traffic dataset, which exhibits possible two major periods—daily ($P = 24$) and weekly ($P = 168$) patterns. In practical multi-period scenarios, we typically use the shorter period as the baseline configuration. Our

Table 8: Sensitivity experiments of period length $P$.

| Method | | MoFo | | +5% | | +15% | | -5% | | -15% | | + Half | | + Double | |
|---|---|---|---|---|---|---|---|---|---|---|---|---|---|---|---|
| Metric | | MSE | MAE | MSE | MAE | MSE | MAE | MSE | MAE | MSE | MAE | MSE | MAE | MSE | MAE |
| ETTm1 | 96 | **0.286** | **0.335** | 0.292 | 0.335 | 0.297 | 0.338 | 0.291 | 0.337 | 0.297 | 0.341 | 0.302 | 0.346 | 0.295 | 0.359 |
| | 192 | **0.320** | **0.363** | 0.329 | 0.368 | 0.336 | 0.370 | 0.333 | 0.369 | 0.344 | 0.386 | 0.335 | 0.379 | 0.339 | 0.388 |
| | 336 | **0.347** | **0.382** | 0.357 | 0.388 | 0.372 | 0.391 | 0.354 | 0.385 | 0.376 | 0.402 | 0.374 | 0.404 | 0.372 | 0.408 |
| | 720 | **0.388** | **0.411** | 0.406 | 0.420 | 0.403 | 0.416 | 0.402 | 0.415 | 0.411 | 0.420 | 0.414 | 0.425 | 0.419 | 0.429 |
| Weather | 96 | **0.141** | **0.186** | 0.143 | 0.192 | 0.155 | 0.206 | 0.157 | 0.221 | 0.146 | 0.188 | 0.152 | 0.199 | 0.157 | 0.206 |
| | 192 | **0.186** | **0.230** | 0.196 | 0.236 | 0.202 | 0.252 | 0.199 | 0.240 | 0.201 | 0.242 | 0.193 | 0.239 | 0.194 | 0.234 |
| | 336 | **0.233** | **0.272** | 0.238 | 0.277 | 0.245 | 0.286 | 0.235 | 0.280 | 0.243 | 0.286 | 0.241 | 0.279 | 0.244 | 0.284 |
| | 720 | **0.312** | **0.331** | 0.335 | 0.358 | 0.323 | 0.358 | 0.332 | 0.352 | 0.335 | 0.355 | 0.348 | 0.363 | 0.342 | 0.355 |
| Traffic | 96 | **0.362** | **0.247** | 0.388 | 0.266 | 0.395 | 0.265 | 0.383 | 0.255 | 0.382 | 0.265 | 0.376 | 0.258 | 0.370 | 0.252 |
| | 192 | **0.379** | **0.254** | 0.397 | 0.275 | 0.408 | 0.287 | 0.398 | 0.284 | 0.409 | 0.288 | 0.380 | 0.262 | 0.388 | 0.275 |
| | 336 | **0.390** | **0.258** | 0.395 | 0.260 | 0.405 | 0.275 | 0.395 | 0.266 | 0.417 | 0.294 | 0.406 | 0.286 | 0.395 | 0.283 |
| | 720 | **0.424** | **0.281** | 0.424 | 0.281 | 0.443 | 0.298 | 0.437 | 0.283 | 0.443 | 0.303 | 0.434 | 0.297 | 0.438 | 0.302 |

experiments show that this choice enables MoFo to capture fine-grained temporal variations while retaining stable performance across longer periods, as shown in Table 9. To explore whether integrating multiple MoFo models could further improve prediction, we introduce three variants: (1) **MoFo-1**, trained with a 24-hour (daily) period (default setting of MoFo); (2) **MoFo-7**, trained with a 7-day (weekly) period; and (3) **Mix-MoFo**, which ensembles both models and combines their outputs through a learnable fusion layer. The results indicate that using the shorter period configuration provides the most precise periodic alignment, while the ensemble model further stabilizes performance under complex multi-period signals. These findings confirm MoFo's robustness and adaptability when modeling time series with heterogeneous or nested periodic patterns.

Table 9: Sensitivity experiments of multiple coexisting periods.

| Method | | MoFo-1 | | Mix-MoFo | | MoFo-7 | |
|---|---|---|---|---|---|---|---|
| Metric | | MSE | MAE | MSE | MAE | MSE | MAE |
| Traffic | 96 | **0.362** | **0.247** | 0.365 | 0.254 | 0.381 | 0.257 |
| | 192 | **0.379** | **0.254** | 0.380 | 0.255 | 0.394 | 0.262 |
| | 336 | **0.390** | **0.258** | 0.395 | 0.263 | 0.409 | 0.272 |
| | 720 | **0.424** | **0.281** | 0.430 | 0.285 | 0.451 | 0.289 |

❸ **Sensitivity to dynamic periods.** To further evaluate robustness of MoFo under complex temporal settings, we conduct experiments on datasets with mixed and time-varying periodicities. To systematically investigate this issue, we construct a synthetic dataset named Mixed-ETT by combining two standard datasets, ETTh1 (period = 24) and ETTm1 (period = 96). Specifically, each dataset is evenly divided into four temporal segments, and these segments are alternately concatenated along the time axis to form the new sequence as Mixed-ETT = $\{\text{ETTh1}[: \frac{1}{4}], \text{ETTm1}[\frac{1}{4} : \frac{1}{2}], \text{ETTh1}[\frac{1}{2} : \frac{3}{4}], \text{ETTm1}[\frac{3}{4} : 1]\}$. This design produces a dataset with alternating periodic structures of 24, 96, 24, 96, mimicking the temporal heterogeneity commonly observed in real applications. The other experimental settings are kept consistent with the main experiments in Section 4. We compare MoFo against two representative baselines, DUET and DLinear. As shown in Table 10, experimental results demonstrate that our straightforward MoFo implementation achieved strong performance. When dealing with multi-period time series, we set the smallest period length as our baseline configuration. The "Mix-MoFo" variant added assumptions regarding additional periods, which increased the risk of overfitting and consequently led to a decline in performance. These results confirm that MoFo can effectively handle dynamic and mixed periodic behaviors without explicit retraining or manual period adjustment.

## D.8 Efficiency Analysis

We compare the computational efficiency of Transformer-based baseline models on the Solar dataset with $L = 96$. As shown in Table 11, MoFo achieves the least MSE among all Transformer-based models, while demonstrating the least parameters number with fastest training speed. All the benefits of MoFo arise from its effective Period-based Discrete Patching strategy, which reduces the complexity of the Transformer to quadratic in relation to the period length, while utilizing only

Table 10: Sensitivity experiments of period-varying synthetic dataset.

| Method | | MoFo | | DUET | | DLinear | |
|---|---|---|---|---|---|---|---|
| Metric | | MSE | MAE | MSE | MAE | MSE | MAE |
| Mixed-ETT | 96 | **0.178** | **0.223** | 0.186 | 0.229 | 0.194 | 0.232 |
| | 192 | 0.185 | **0.232** | **0.184** | 0.241 | 0.190 | 0.247 |
| | 336 | **0.180** | **0.237** | 0.187 | 0.247 | 0.196 | 0.256 |
| | 720 | **0.191** | **0.253** | 0.218 | 0.278 | 0.219 | 0.282 |

a single layer of the Transformer, which is sufficient. Similarity, the competitive baseline DUET with dual clusting strategy on both temporal dimension and channel dimension exhibits over $14\times$ increase in parameters number, $4\times$ higher FLOPs, and slower training speed. Compared to PatchTST, a classic Transformer-based model that employs a successive patching strategy on the time series, MoFo reduces the number of MACs by more than $80\times$, accelerates the training speed by over $10\times$, and significantly decreases both the parameter requirements and memory usage. This is primarily because PatchTST stacks multiple layers of the Transformer, which is necessary for its architecture, yet lacks a reasonable positional encoding for time series data. Although iTransformer, FEDformer, and Informer have fewer computations than MoFo, they require longer training times as well as larger parameters, and their performance lags behind MoFo by up to $1.8\times$ since their complexity architectures.

Table 11: Efficiency comparison of MoFo and SOTA baselines with $L = 720$ in Traffic dataset. All results of each model are under the optimal hyperparameters for fair comparison. Parameters: All learnable parameters requiring gradient descent. MACs: multiply–accumulate operations. FLOPs: floating point operations. M: Million ($10^6$). B: Billion ($10^9$). T: Trillion ($10^{12}$). MB: Megabyte. s: Second. $\uparrow$ indicates the relative percentage increasing regarding MoFo and $\downarrow$ indicates the relative percentage decreasing.

| | Models | MSE | # Parameters | # MACs | # FLOPs | Memory Usage | Epoch Time |
|---|---|---|---|---|---|---|---|
| Solar [$L = 96$] | Informer | 0.368 ↑117.75% | 2.26 M ↑527.78% | 7.13 B ↓2.72% | 7.18 B ↓38.58% | 852 MB ↓75.36% | 58 s ↑222.22% |
| | Stationary | 0.365 ↑115.97% | 11.2 M ↑3011.11% | 39.56 B ↑439.70% | 41.32 B ↑253.46% | 1,710 MB ↓50.54% | 24 s ↑33.33% |
| | Triformer | 0.225 ↑33.14% | 1.62 M ↑350.00% | 33.56 B ↑357.84% | 38.45 B ↑228.91% | 11,714 MB ↑238.75% | 147 s ↑716.67% |
| | FEDformer | 0.485 ↑186.98% | 3.63 M ↑908.33% | 1.86 B ↓74.62% | 1.52 B ↓86.99% | 858 MB ↓75.19% | 204 s ↑1033.33% |
| | Crossformer | 0.183 ↑8.28% | 3.82 M ↑961.11% | 159.3 B ↑2073.26% | 166.1 B ↑1320.87% | 10,698 MB ↑209.37% | 101 s ↑461.11% |
| | PatchTST | 0.170 ↑0.59% | 2.62 M ↑627.78% | 607.4 B ↑8186.49% | 644.9 B ↑5416.68% | 29,726 MB ↑759.63% | 215 s ↑1094.44% |
| | Pathformer | 0.218 ↑28.99% | 5.72 M ↑1488.89% | 24.84 B ↑238.88% | 27.99 B ↑139.44% | 12,754 MB ↑268.83% | 614 s ↑3311.11% |
| | iTransformer | 0.190 ↑12.42% | 0.51 M ↑41.67% | 2.32 B ↓68.34% | 2.66 B ↓77.24% | 788 MB ↓77.21% | 18 s ↑0.00% |
| | PDF | 0.181 ↑7.10% | 5.82 M ↑1516.67% | 204.1 B ↑2684.45% | 208.0 B ↑16790.30% | 5,616 MB ↑62.41% | 25 s ↑38.89% |
| | DUET | 0.169 ↑0.00% | 5.64 M ↑1466.67% | 19.23 B ↑162.35% | 60.88 B ↑420.79% | 4,422 MB ↑27.88% | 35 s ↑94.44% |
| | **MoFo** | **0.169** | **0.36** M | **7.33** B | **11.69** B | **3,458** MB | **18** s |

# E  Discussion and Future Work

MoFo performs discrete patching based on periodic position, grouping the most correlevant time steps within the same patch for prioritized interaction. However, our periodic positional embedding provided by the Regulated Relaxation Function is currently only applicable to vanilla self-attention mechanisms, which exhibit quadratic complexity (although our specific implementation's complexity is quadratic with respect to the period length $P$ rather than the sequence length $T$ with $P \ll T$). Exploring how to adapt this method to attention mechanisms with linear complexity is a direction worthy of future investigation. Simultaneously, applying the core ideas of MoFo within time series LLM foundational models to empower their development in time series learning is another promising avenue for subsequent research.

