# OpenReview forum: "MoFo: Empowering Long-term Time Series Forecasting with Periodic Pattern Modeling"
_NeurIPS.cc/2025/Conference — NeurIPS 2025 poster_

### Official Review · Reviewer_Q3nE · 2025-06-05

**Clarity:** 3
**Significance:** 3
**Originality:** 3
**Rating:** 5
**Confidence:** 4

**Summary:**

The paper proposes the MoFo framework for long-term time series forecasting, which models periodicity as the correlation of period-aligned time steps and the trend of period-offset time steps. By designing a periodic structural patch and a periodic-aware modulator, the framework captures periodic patterns effectively. Experiments demonstrate that it outperforms 17 baselines, achieving both a 14x improvement in memory efficiency and a 10x faster training speed.

**Questions:**

The authors can refer to the weakness listed above.

**Ethical Concerns:**

["NO or VERY MINOR ethics concerns only"]

**Final Justification:**

The paper proposes the MoFo framework for long-term time series forecasting, which models periodicity as the correlation of period-aligned time steps and the trend of period-offset time steps. By designing a periodic structural patch and a periodic-aware modulator, the framework captures periodic patterns effectively. Experiments demonstrate that it outperforms 17 baselines, achieving both a 14x improvement in memory efficiency and a 10x faster training speed. The author's response has largely addressed my concerns. Considering the significance of MoFo for long-term time series forecasting, I have raised my score to 5.

**Limitations:**

The authors can refer to the weakness listed above.

**Paper Formatting Concerns:**

The authors can refer to the weakness listed above.

**Quality:**

3

**Strengths And Weaknesses:**

**Strengths**

1. Modeling periodicity as the correlation of period-aligned time steps and the trend of period-offset time steps provides a novel modeling perspective for long-term time series forecasting.
2. MoFo demonstrates competitive predictive performance compared to 17 state-of-the-art baselines on multiple popular benchmark datasets, validating the effectiveness of the proposed method.
3. The model's computational complexity is related to the period length rather than the input sequence length, achieving up to a 14$\times$ improvement in memory efficiency and a 10$\times$ speedup in training.

**Weaknesses**

1. Presentation Issues
   - Grammatical errors. For example, there is a subject-verb agreement issue on line 182 regarding singular and plural forms.
   - The names of all designed modules should be in lowercase.
2. Clarifications Needed
   - The description of the patch-based structured input is unclear, especially the padding strategy. Using data from the last period for padding may introduce potential information redundancy —could this affect model generalization?
   - How to determine the cycle length? What does formula H mean in Eq.10?
3. Baselines
   - How many layers of Transformer did the author use? Did the author perform a sensitivity analysis on this parameter?
   - A hyperparameter study on the period length is suggested, for example, using double the default period length, to evaluate its impact on performance.

---

> ### Author Rebuttal · Authors · 2025-07-31
>
> Dear Reviewer Q3nE,
>
> Thank you for your kind words. Your feedback and guidance have been instrumental in improving our paper, and we sincerely appreciate your support and constructive suggestions. Next, we will respond to your concerns point by point.
>
> ## **W1. Presentation Typos**
>
> We sincerely appreciate your constructive feedback and will thoroughly revise and proof-read the manuscript in the next version to address every point you raised.
>
> ## **W2.1. Padding Strategy**
>
> Our padding strategy is already described in detail in **Section 3.1, lines 109–119** and formalized in **Equation (1)** of the manuscript. In fact, Given an input sequence $\mathbf{X} = [\mathbf{x}_1, …, \mathbf{x}_T]^\top$, we start from the current time step $\mathbf{x}_T$ and move past to delineate periods of length $P$. Any prefix that does not form a complete period is left-padded with the leftmost elements of the nearest full period on its right; this yields a padded input length of $\lceil T / P\rceil * P$ and exactly $\lceil T / P\rceil$ periods.
>
> To evaluate the impact of our strategy, we introduce a variant "+ zero padding" that pads the value with zeros and conpare on ETTm1 and ETTm2 datasets. The results are shown below and the best-performing metric for each experiment is bolded.
>
>
> | Dataset | Method  | MoFo              | + zeros padding           |
> |---------|--------|-------------------|---------------------------|
> |         | Prediction Window    |       MSE, MAE    |    MSE, MAE               |
> | ETTm1   | 96     | **0.286, 0.335**  | 0.292, 0.345              |
> |         | 192    | **0.320, 0.363**  | 0.328, 0.372              |
> |         | 336    | **0.347, 0.382**  | 0.348, 0.386              |
> |         | 720    | **0.388, 0.411**  | 0.401, 0.435              |
> | ETTm2   | 96     | **0.155, 0.240**  | 0.156, 0.247              |
> |         | 192    | **0.211, 0.283**  | 0.215, 0.293              |
> |         | 336    | **0.258, 0.314**  | 0.261, 0.318              |
> |         | 720    | **0.342, 0.368**  | 0.345, 0.372              |
>
>
> Our padding strategy consistently outperforms zero padding, demonstrating that re-using the immediately preceding historical pattern enhances generalization rather than introducing redundancy.
>
>
> ## **W2.2.1. Determine the cycle length**
>
> We determine the period length by calculating the distance between the peaks of the autocorrelation coefficients of the time series data on the training set. The specific method for this calculating can be found in **Appendix A.2**.
>
>
> ## **W2.2.2. What is $\mathcal{H}$ ?**
>
> We apologize for omitting the precise definition. In our notation, $\mathcal{H}$ denotes the Heaviside step function satisfying
> $$
> \mathcal{H}(s) = 0 \text{ if } s < 0; \mathcal{H}(s) = 1 \text{ if } s > 0.
> $$
> We will add this clarification to the next version.
>
> ## **W3.1. Transformer layers in MoFo**
>
> **MoFo attains best performance with only one Transformer layer**. The corresponding ablation is provided in **Section 4.6** and **Figure 5** of the manuscript, where we systematically vary the number of Transformer blocks. Across all settings, stacking additional layers yields negligible to zero gains, confirming that a single layer already saturates the model’s capacity.
>
> ## **W3.2. Period length of MoFo**
>
> Following your suggestion, we designed two variants—**"+ Half"** and **"+ Double"**—to investigate MoFo’s sensitivity to the chosen the half and the double period length. As shown the below Table, the results on three diverse datasets with different period reveal that a well-calibrated period is essential: an inappropriate choice disrupts the model’s ability to capture the intrinsic periodicity of the time series, whereas the original setting preserves this structure and retains optimal performance.
>
>
> | Dataset| Method   | MoFo                | + Half              | + Double            |
> |----------|--------|---------------------|---------------------|---------------------|
> |          | Prediction Window    |       MSE, MAE      |     MSE, MAE        |     MSE, MAE        |
> | ETTm1    | 96     | **0.289, 0.336**    | 0.302, 0.346        | 0.295, 0.359        |
> | | 192    | **0.326, 0.367**    | 0.335, 0.379        | 0.339, 0.388        |
> |          | 336    | **0.361, 0.391**    | 0.374, 0.404        | 0.372, 0.408        |
> |          | 720    | **0.405, 0.419**    | 0.414, 0.425        | 0.419, 0.429        |
> | Weather  | 96     | **0.141, 0.186**    | 0.152, 0.199        | 0.157, 0.206        |
> | | 192    | **0.186, 0.230**    | 0.193, 0.239        | 0.194, 0.234        |
> |          | 336    | **0.233, 0.272**    | 0.241, 0.279        | 0.244, 0.284        |
> |          | 720    | **0.312, 0.331**    | 0.348, 0.363        | 0.342, 0.355        |
> | Traffic  | 96     | **0.362, 0.247**    | 0.376, 0.258        | 0.370, 0.252        |
> | | 192    | **0.379, 0.254**    | 0.380, 0.262        | 0.388, 0.275        |
> |          | 336    | **0.390, 0.258**    | 0.406, 0.286        | 0.395, 0.283        |
> |          | 720    | **0.424, 0.281**    | 0.434, 0.297        | 0.438, 0.302        |

---

> > ### Comment · Reviewer_Q3nE · 2025-08-01
> >
> > Thanks for the your reply. This has largely addressed my concerns. Considering the importance of MoFo for long-term time series forecasting, I have raised my score to 5.

---

> > > ### Author Response · Authors · 2025-08-04
> > >
> > > Dear Reviewer Q3nE,
> > >
> > > We sincerely value your positive feedback and the improvement in our score. Your insightful guidance has been immensely helpful, and we will carefully revise our work to incorporate all of your suggestions.
> > >
> > > With heartfelt thanks,
> > >
> > > Authors.

---

### Official Review · Reviewer_XTBU · 2025-06-29

**Clarity:** 4
**Significance:** 3
**Originality:** 3
**Rating:** 5
**Confidence:** 3

**Summary:**

This paper introduces MoFo, a Transformer-based approach for long-term sequence forecasting. The method employs a periodicity-structured patching strategy along with a periodicity-aware modulator, inspired by theoretical mathematical insights. Extensive experimental results demonstrate that MoFo achieves highly competitive performance and superior efficiency.

**Questions:**

Please kindly see the Weaknesses above.

**Ethical Concerns:**

["NO or VERY MINOR ethics concerns only"]

**Final Justification:**

Since authors have targeted and addressed my questions in rebuttal I keep the score conservatively.

**Limitations:**

YES

**Quality:**

3

**Strengths And Weaknesses:**

### Strength:
S1. The paper has a strong and clearly articulated motivation. The authors have effectively identified and explained the gap in existing work on time series forecasting.

S2. The proposed method is novel, incorporating a periodicity-aware modulator and an input representation strategy that emphasize the modeling of temporal periodicity. And the approach is driven by theoretical mathematical insights.

S3. Extensive experiments validate the model’s effectiveness, showcasing significant performance improvements and approximately 15× efficiency gains.

S4.  The paper is well-written and well-structured, and The mathematical formulas are presented in a clear and explicit manner.

### Weakness:
W1. Some time series forecasting benchmarks need to be supplemented. For example, how does the model compare to FITS and SparseTF models?

W2. Why does the time complexity of MoFo remain stable as the input time series becomes longer?

W3. What does MACs mean in Table 3?  What is the purpose of this indicator?

---

> ### Author Rebuttal · Authors · 2025-07-31
>
> We appreciate your guidance—your suggestions have helped us produce a more precise, readable, and broadly accessible manuscript.
>
> ## **W1. More Baselines**
>
> Thank you for your valuable suggestion. The FITS [5] model is already included in Table 4 of Appendix E.3. To further address your concern, we have conducted additional experiments comparing FITS [5] and SparseTF [6] on the ETTm1 and ETTm2 datasets; the results are reported below. For clarity, the best performance in each experiment is highlighted in bold. Across all evaluations, MoFo consistently achieves the most competitive results, demonstrating superior performance.
>
> |  Dataset| Method| MoFo              | FITS            | SparseTSF            |
> |---------|--------|-------------------|-----------------|----------------------|
> |         | Prediction Window    |       MSE, MAE    |        MSE, MAE |        MSE, MAE      |
> | ETTm1   | 96     | **0.286**, **0.335**  | 0.303, 0.345    | 0.316, 0.355         |
> |         | 192    | **0.320**, **0.363**  | 0.337, 0.365    | 0.348, 0.376         |
> |         | 336    | **0.347**, **0.382**  | 0.368, 0.384    | 0.373, 0.387         |
> |         | 720    | **0.388**, **0.411**  | 0.420, 0.413    | 0.434, 0.422         |
> | ETTm2   | 96     | **0.155**, **0.240**  | 0.165, 0.254    | 0.166, 0.256         |
> |         | 192    | **0.211**, **0.283**  | 0.219, 0.291    | 0.220, 0.292         |
> |         | 336    | **0.258**, **0.314**  | 0.272, 0.326    | 0.273, 0.327         |
> |         | 720    | **0.342**, **0.368**  | 0.359, 0.381    | 0.361, 0.381         |
>
> Furthermore, as requested by Reviewer 7GM5, we have added comparisons with the TimesNet [1], WITRAN [2], PGN[3], and TimeKAN[4] models. Please refer to our response to W1 for details.
>
> ## **W2. Why does the time complexity of MoFo remain stable as the input time series becomes longer?**
>
> In MoFo, due to the unique patch strategy, the number of patches is equal to the period length and thus remains constant. Each patch contains exactly $\lceil T / P\rceil$ elements, where $T$ is the input sequence length and $P$ is the period length. Since the time complexity depends solely on the number of patches (i.e., the period length), MoFo is not sensitive to increases in the input sequence length.
>
> ## **W3. What is MACs**
>
> MACs (Multiply–Accumulate operations) quantify the total number of fused multiply-add operations executed during a single forward pass. This metric serves as a hardware-agnostic proxy for computational cost: fewer MACs generally translate to lower latency and reduced energy consumption on both CPUs and accelerators. By reporting MACs alongside accuracy, we provide a clear, hardware-independent view of the efficiency–performance trade-off offered by each model.
>
> Ref:
>
> [1] Wu, H., Hu, T., Liu, Y., Zhou, H., Wang, J., & Long, M. (2023). Timesnet: Temporal 2d-variation modeling for general time series analysis. In The Eleventh International Conference on Learning Representations.
>
> [2] Jia, Y., Lin, Y., Hao, X., Lin, Y., Guo, S., & Wan, H. (2023). WITRAN: Water-wave Information Transmission and Recurrent Acceleration Network for Long-range Time Series Forecasting. In Thirty-seventh Annual Conference on Neural Information Processing Systems.
>
> [3] Jia, Y., Lin, Y., Yu, J., Wang, S., Liu, T., & Wan, H. (2024). PGN: The RNN's New Successor is Effective for Long-Range Time Series Forecasting. In The Thirty-eighth Annual Conference on Neural Information Processing Systems.
>
> [4] Huang, S., Zhao, Z., Li, C., & Bai, L. (2025). TimeKAN: KAN-based Frequency Decomposition Learning Architecture for Long-term Time Series Forecasting. In The Thirteenth International Conference on Learning Representations.
>
> [5] Xu, Z., Zeng, A., & Xu, Q. (2023). FITS: Modeling time series with $10 k $ parameters. arXiv preprint arXiv:2307.03756.
>
> [6] Lin, S., Lin, W., Wu, W., Chen, H., & Yang, J. (2024, July). SparseTSF: Modeling Long-term Time Series Forecasting with* 1k* Parameters. In International Conference on Machine Learning (pp. 30211-30226). PMLR.

---

> ### Comment · Reviewer_XTBU · 2025-08-05
>
> Thank you for your detailed response and the additional experimental results. Regarding the additional experiments and analysis you provided, I appreciate the efforts of the authors in the rebuttal phase, and as a result, I conservatively keep the rating. I will keep tracking the discussion with 7gm5, who provide actionable discussions.

---

> ### Author Response · Authors · 2025-08-09
>
> Dear Reviewer XTBU,
>
> Thank you for your insightful feedback during the review phase, and we appreciate your support of our submission and your attention during the discussion phase between us and Reviewer 7gm5.
>
> In response to Reviewer 7gm5's actionable discussions on two points: (1) the specific impact of model layers, and (2) evaluation on larger-scale datasets, we have conducted corresponding experiments and have addressed Reviewer 7gm5's concerns, as detailed in our subsequent response to Reviewer 7gm5 at [https://openreview.net/forum?id=sbvLts2HqR&noteId=fAooVNcIEf].
>
> Thank you once again for your valuable feedback. Your thorough review has been truly encouraging for us.
>
> Best regards,
>
> The Authors of Submission 2224

---

### Official Review · Reviewer_7GM5 · 2025-07-02

**Clarity:** 3
**Significance:** 3
**Originality:** 2
**Rating:** 4
**Confidence:** 5

**Summary:**

This paper proposes MoFo, a Transformer-based forecasting method that models time series in a two-dimensional structure by leveraging their periodic patterns. The authors conduct correlation-based experiments in an attempt to validate the effectiveness of the proposed approach.

**Questions:**

Is MoFo still effective on time series data without clear periodicity? Has the method been explored on such datasets, for example, the Human Connectome Project (HCP), which involves non-periodic time series? I would like to see more discussion on the applicability and performance of the proposed method in non-periodic scenarios.

**Ethical Concerns:**

["NO or VERY MINOR ethics concerns only"]

**Final Justification:**

The authors have addressed my concern. Therefore, I am raising my score to 4.

**Limitations:**

The authors mentioned limitations regarding computational complexity.

**Paper Formatting Concerns:**

It seems that the NeurIPS 2025 submission guidelines explicitly state that tables should not contain vertical rules, yet vertical rules are included in Table 2 and Table 4 of the paper.

**Quality:**

3

**Strengths And Weaknesses:**

Strengths:

S1: The method for handling periodicity in the paper is reasonable and demonstrates a certain degree of novelty.

S2: The figures and tables throughout the paper are clear, and the presentation is well-structured.

Weaknesses:

W1: The paper lacks comparisons with important baseline methods, especially since the proposed method also relies on 2D modeling. Therefore, it should be compared with previous classic SOTA methods that also use 2D modeling, such as TimesNet[1] (CNN-based), WITRAN[2] (RNN-based), and TPGN[3] (PGN-based, where PGN is a new paradigm based on RNN). Moreover, in the related work section, these key methods are not addressed, and other emerging approaches, such as TimeKAN[4] (KAN-based, where KAN is a new paradigm based on Linear), are also omitted.

[1] Wu, H., Hu, T., Liu, Y., Zhou, H., Wang, J., & Long, M. (2023). Timesnet: Temporal 2d-variation modeling for general time series analysis. In The Eleventh International Conference on Learning Representations.

[2] Jia, Y., Lin, Y., Hao, X., Lin, Y., Guo, S., & Wan, H. (2023). WITRAN: Water-wave Information Transmission and Recurrent Acceleration Network for Long-range Time Series Forecasting. In Thirty-seventh Annual Conference on Neural Information Processing Systems.

[3] Jia, Y., Lin, Y., Yu, J., Wang, S., Liu, T., & Wan, H. (2024). PGN: The RNN's New Successor is Effective for Long-Range Time Series Forecasting. In The Thirty-eighth Annual Conference on Neural Information Processing Systems.

[4] Huang, S., Zhao, Z., Li, C., & Bai, L. (2025). TimeKAN: KAN-based Frequency Decomposition Learning Architecture for Long-term Time Series Forecasting. In The Thirteenth International Conference on Learning Representations.

W2: The experimental setup contains clear flaws. Although the authors claim to follow configurations from previous studies, the actual design lacks fairness and rigor.

(1) Improper handling of historical look-back length

(1.1) Different look-back lengths may alter the data splitting process, thereby affecting model training. This represents a lack of control over experimental variables.

(1.2) Different models exhibit varying sensitivity to look-back lengths, so treating it uniformly as a hyperparameter is clearly inappropriate. A more fair and rigorous approach would be to predefine several common look-back lengths and treat them as separate experimental groups, enabling a more systematic comparison of how each method performs under different historical horizons.

(2) Limited hyperparameter search space
The TFB paper cited by the authors only expands to 8 parameter configurations. Taking Transformers as an example, even commonly used parameters like d_model, e_layer, and d_layer are barely covered—let alone other hyperparameters. Such a narrow search space makes it difficult to draw convincing conclusions.

(3) Why an extensive hyperparameter search is necessary

(3.1) Even with identical random seeds and hyperparameter values, experimental results may still vary across hardware platforms.

(3.2) Different models have different levels of sensitivity to hyperparameters. When the experimental platform changes, directly reusing published settings cannot guarantee that baseline models operate at their optimal capacity, nor that results are consistent with those reported originally.

Therefore, to ensure a fair comparison, it is essential to perform a fresh hyperparameter search for all models within a unified and sufficiently broad search space by selecting optimal configurations on the validation set and then evaluating and reporting results on the test set. While models with more hyperparameters may lead to a larger search space, this process is critical for ensuring fairness and rigor. Only after eliminating platform differences and hyperparameter biases, and ensuring that all models perform at their best, can a trustworthy and meaningful comparison be made.

Furthermore, the authors do not compare their approach with the baselines mentioned in W1. Consequently, the current experiments are insufficient to convincingly establish the effectiveness of the proposed method.

W3: The efficiency experiments likewise omit comparisons with the baselines listed in W1, as well as with FITS and DLinear. In addition, the paper does not specify how the hyperparameters were chosen for these tests. It must be emphasized that efficiency studies are intended to measure the resource consumption inherent to each model architecture; hence, all experimental variables should be controlled to ensure a fair comparison. In particular, shared hyperparameters should be kept identical so as to eliminate confounding effects, such as one method being run with very small settings while another is evaluated with much larger ones.

---

> ### Author Rebuttal · Authors · 2025-07-31
>
> Thank you very much for your time and effort; they are crucial for improving the quality of our manuscript.
>
> ----
>
> ## **W1. More baselines**
>
> We have already compared TimesNet in Table 4. Next, we added comparative experiments with PGN, TimeKAN, and WITRAN. The performance results are shown in the table below.
>
> | Datasetr | Method | MoFo | TimeKAN  | PGN  | TimesNet  | WITRAN  |
> |--------|--------|-----------------|--------------------|----------------|---------------------|-------------------|
> |          | Prediction Window       |       MSE,  MAE |       MSE, MAE |          MSE, MAE  |          MSE, MAE  |          MSE, MAE  |
> | ETTh1  | 96     | **0.360**, **0.389**| 0.369, 0.396       | 0.402, 0.407   | 0.389, 0.412        | 0.660, 0.627      |
> |        | 192    | **0.397**, **0.413**| 0.402, 0.417       | 0.446, 0.433   | 0.440, 0.443        | 0.844, 0.723      |
> |        | 336    | **0.407**, **0.424**| 0.419, 0.430       | 0.494, 0.456   | 0.523, 0.487        | 0.967, 0.766      |
> |        | 720    | 0.447, **0.454**| **0.442**, 0.463       | 0.490, 0.478   | 0.521, 0.495        | 0.965, 0.768      |
> | ETTh2  | 96     | 0.273, **0.334**| **0.279**, 0.343       | 0.323, 0.358   | 0.334, 0.370        | 0.601, 1.285      |
> |        | 192    | 0.327, **0.373**| **0.326**, 0.380       | 0.426, 0.420   | 0.404, 0.413        | 4.193, 1.611      |
> |        | 336    | **0.361**, **0.405**| 0.368, 0.411       | 0.452, 0.457   | 0.389, 0.435        | 5.036, 1.819      |
> |        | 720    | **0.379**, **0.425**| 0.419, 0.443       | 0.438, 0.449   | 0.434, 0.448        | 4.556, 1.817      |
>
> > **Fairness statement**. Although the TFB-benchmark setting we adopted may differ from your expectations, we would like to clarify that all models are run under the same experimental environment and unified configuration to ensure basic fairness in evaluation. We chose to conduct experiments on the latest TFB mainly because it has been extensively validated and is academically recognized—the benchmark was nominated for the Best Paper Award at VLDB 2024 and has been cited over 62 times on Google Scholar.
>
> ----
>
> ## **W2. Historical look-back length**
>
> Please allow us to clarify TFB’s approach regarding historical window selection. According to the TFB GitHub guidelines, multiple look-back window lengths are preset in the experiments, covering typical settings commonly used in time series forecasting tasks. For each model, tests are conducted independently on all candidate historical window lengths, and the best-performing result is ultimately reported.
>
> We believe this approach does not introduce fairness issues; on the contrary, it is fairer to models with varying sensitivities. Forcing all models to use exactly the same look-back length might disadvantage those that are particularly sensitive to this parameter, causing them to be unfairly regarded as “underperforming”. **In contrast, as long as the device conditions and the amount of historical data are consistent, allowing each model to flexibly choose its optimal historical window according to its structural characteristics both ensures a fairer evaluation of predictive performance and better reflects the flexible practices found in real-world applications.**
>
> Despite having a different perspective, in response to your suggestion, we conducted experiments using several commonly adopted predefined look-back window lengths. Specifically, for forecasting with a future window length of 720, we used {96, 336, 512} as the look-back windows. Due to time constraints, partial experimental results are shown below.
>
> || Metric | MoFo  | TimeKAN  | FITS  | PGN   | TimesNet  | WITRAN  | DLinear |
> |-------|--------|-----------------------|-------------------------|---------------------|---------------------|-------------------------|--------------------|---------------------|
> |          |   Look-back Window    |       MSE,  MAE |       MSE, MAE |          MSE, MAE  |          MSE, MAE  |          MSE, MAE  |          MSE, MAE  |          MSE, MAE  |
> | ETTh1 | 96     | **0.447**, **0.454**         | 0.463, 0.470           | 0.547, 0.518         | 0.490, 0.478         | 0.521, 0.495         | 0.965, 0.768       | 0.515, 0.511         |
> |       | 336    | **0.459**, **0.469**         | 0.509, 0.490           | 0.475, 0.487         | 0.522, 0.515         | 0.542, 0.519         | 0.976, 0.771       | 0.471, 0.493         |
> |       | 512    | 0.443, 0.463         | 0.442, 0.463           | **0.435**, **0.458**         | 0.692, 0.598         | 0.560, 0.531         | 1.034, 0.802       | 0.464, 0.488         |
> | ETTh2 | 96     | **0.416**, **0.442**         | 0.438, 0.451           | 0.439, 0.452         | 0.438, 0.450         | 0.434, 0.448         | 4.556, 1.817       | 0.650, 0.571         |
> |       | 336    | **0.393**, **0.428**         | 0.420, 0.449           | 0.397, 0.431         | 0.466, 0.474         | 0.472, 0.480         | 4.272, 1.745       | 0.704, 0.597         |
> |       | 512    | **0.379**, **0.425**         | 0.408, 0.443           | 0.382, 0.425         | 0.440, 0.465         | 0.480, 0.468         | 4.668, 1.787       | 0.786, 0.623         |
>
> ----
>
> ## **W3. Hyperparameter search**
>
> (1) According to the TFB guidelines, the authors follow the hyperparameters specified in the original papers. And they further performed additional hyperparameter searches on 16 configuration sets to achieve a more comprehensive evaluation, which covers the key hyperparameters sufficiently.
>
> (2) TFB has fully open-sourced the necessary environment configuration, including detailed version information, along with a fixed running protocol, enabling users to reproduce the same performance.
>
> (3) Please refer to W1.
>
> ----
>
> ## **W4. Complexity**
>
> **Our efficiency comparison, as stated in the caption of Table 3, is conducted under the condition that each model achieves its optimal predictive performance—a standard and widely accepted practice in academic research. Since the ultimate goal of time series forecasting is to enhance prediction accuracy, efficiency comparisons that consider performance are both meaningful and practical.** For instance, if all models are configured with extreme parameter settings, like compressing the feature dimension to 1, computational efficiency might seem significantly improved, but prediction performance would suffer greatly. Therefore, a fair and informative efficiency evaluation must be conducted at comparable performance levels to ensure both validity and practical significance.
>
> In response to your suggestion, we used the ETTh1 dataset as an example and optimized this configuration for the best performance, while keeping certain hyperparameters such as batch size and learning rate fixed. We report the efficiency for fixed batch sizes (specifically 4 and 128), learning rate 1e-4 and look-back window 512, as well as for each model under its best-performing setting. Please refer to the table below for details (K: Kilo-$10^3$, M: Million-$10^6$). We observe that our model achieves significant performance improvements with only a slight increase in complexity.
>
> | ETTh1       | Model   | MSE    | MAE    | Memory   | # Para   | # MACs      | # FLOPs      |
> |-------------|---------|--------|--------|----------|----------|-------------|--------------|
> | bs=4        | MoFo    | **0.372**  | **0.399**  | 542 MB   | 33.0 K   | 2.5 M   | 2.9 M    |
> | | FITS    | 0.381  | 0.406  | 540 MB   | 24,0 K | 0.7 M     | 0.7 M      |
> |  | DLinear | 0.387  | 0.415  | 532 MB   | 98,5 K   | 2.8 M   | 2.8 M    |
> | bs=128 | MoFo    | **0.380**  | **0.390**  | 636 MB   | 22.4 K   | 42.3 M  | 50.6 M   |
> | | FITS    | 0.492  | 0.478  | 584 MB   | 24,0 K   | 21.4 M  | 21.4 M   |
> |  |   DLinear | 0.389  | 0.418  | 558 MB   | 98.5 K   | 88.5 M  | 88.1 M   |
>
> ----
>
> ## **Q1.**
>
> The Human Connectome Project (HCP) is **a large-scale dataset with over 1TB of data**, requiring **mandatory institutional approval for access**. Due to time constraints, we were unable to use and study it during the rebuttal period. To address your concerns, we opted to utilize the Influenza-Like Illness (ILI)  dataset for evaluation instead. The ILI dataset, published by the US Centers for Disease Control and Prevention (CDC) from 2002 to 2021, comprises health-related data that record the weekly proportion of ILI patients relative to the total number of visits, without any obvious seasonality. We predict future values for four windows: $L\in$ {24,36,48,60}.
>
> To emphasize our modeling of non-periodic time series, we follow the approach of TimesNet by computing the main pseudo-period as the reciprocal of the dominant frequency obtained via Fast Fourier Transform (FFT) on each input time series. Although MoFo was designed with periodic patterns in mind, the results below demonstrate that it still achieves remarkably high accuracy on non-periodic data, highlighting its strong capability in capturing complex temporal dynamics even in the absence of clear periodicity.
>
> | | MoFo | TimeKAN | FITS | PGN  | TimesNet  | WITRAN |
> |----------|-----------------|--------------------|-----------------|---------------|---------------------|-------------------|
> |    $L$      |       MSE,  MAE |       MSE, MAE |          MSE, MAE  |          MSE, MAE  |          MSE, MAE  |          MSE, MAE  |          MSE, MAE  |
> | 24       | **2.113**, **0.927**    | 2.176, 0.928       | 2.182, 1.002    | 2.255, 0.936  | 2.131, 0.958        | 3.486, 1.218      | 2.208, 1.031       |
> | 36       | **1.952**, **0.924**    | 2.166, 0.993       | 2.330, 1.051    | 2.132, 0.940  | 2.612, 0.974        | 5.129, 1.596      | 2.032, 0.981       |
> | 48       | **1.714**, **0.824**    | 2.011, 0.928       | 2.761, 1.184    | 2.182, 0.944  | 1.916, 0.897        | 3.862, 1.332      | 2.209, 1.063       |
> | 60       | **1.800**, 0.906    | 2.010, 0.967       | 2.929, 1.217    | 2.169, 0.940  | 1.995, **0.905**        | 4.812, 1.543      | 2.292, 1.086       |

---

> > ### Comment · Reviewer_7GM5 · 2025-08-02
> >
> > Thank you for your response. I noticed that additional experiments have been provided, which seem to demonstrate the effectiveness of the MoFo method from a different perspective. In addition, supplementary experiments on non-periodic data have been included to support the method’s broader applicability. My main concerns have been addressed, and I am therefore willing to adjust my score to a positive rating.
> >
> > However, the following points would benefit from further clarification:
> >
> > (1) The specific impact of model layers on performance;
> >
> > (2) Regarding efficiency, it would be beneficial to include experiments on larger-scale datasets to allow for a more comprehensive evaluation.

---

> > > ### Author Response · Authors · 2025-08-03
> > >
> > > We sincerely appreciate your positive response. Moving forward, we will conduct additional experiments to ensure your satisfaction. Once again, we deeply thank you for improving the evaluation to a positive score.

---

> ### Author Response · Authors · 2025-08-06
>
> Thank you very much for your response. As the rebuttal period is nearing its end, we have done our utmost to address your concerns and ensure your satisfaction.
>
> ## **1. Model Layers**
>
> #### **1.1. Model Layer of MoFo**
> MoFo attains best performance with only one Transformer layer. The corresponding ablation is provided in **Section 4.6 and Figure 5** of the submission, where we systematically vary the number of Transformer blocks. In all configurations, adding more layers does not provide additional performance gains.
>
> #### **1.2. Model Layer of Baselines**
>
> To further address your concerns regarding the baseline model layer parameters, we conducted an investigation using DUET, TimeKAN, and TimesNet. Due to time constraints, we were only able to perform hyperparameter experiments on a single dataset, ETTh1, with prediction lengths of 96, 192, and 336. The optimal layer choices for each prediction length are listed in parentheses. The experimental results are as follows:
>
> |Model Layer| 1        | 2        | 3        | 4        |
> |----------|----------|----------|----------|----------|
> |**DUET**  | MSE, MAE | MSE, MAE | MSE, MAE | MSE, MAE |
> | 96  | **0.352**, **0.384** | 0.358, 0.387 | 0.371, 0.400 | 0.368, 0.396 |
> | 192 | **0.398**, **0.409** | 0.402, 0.413 | 0.410, 0.418 | 0.408, 0.418 |
> | 336| 0.414, **0.426** | **0.413**, 0.430 | 0.413, 0.432 | 0.417, 0.436 |
>
>
> |Model Layer| 1        | 2        | 3        | 4        |
> |----------|----------|----------|----------|----------|
> |**TimeKAN**| MSE, MAE | MSE, MAE | MSE, MAE | MSE, MAE |
> | 96  | 0.375, 0.402 | **0.369**, **0.396** | 0.378, 0.402 | 0.376, 0.401 |
> | 192 | 0.415, 0.420 | **0.402**, **0.417** | 0.412, 0.427 | 0.416, 0.427 |
> | 336| 0.434, 0.437 | **0.419**, **0.430** | 0.434, 0.436 | 0.435, 0.434 |
>
>
> |Model Layer| 1        | 2        | 3        | 4        |
> |----------|----------|----------|----------|----------|
> |**TimesNet**| MSE, MAE | MSE, MAE | MSE, MAE | MSE, MAE |
> | 96 | 0.412, 0.429 | **0.389**, **0.412** | 0.403, 0.421 | 0.418, 0.430 |
> | 192 | 0.485, 0.470 | **0.440**,**0.443** | 0.475, 0.462 | 0.470, 0.462 |
> | 336 | 0.530, 0.493 | **0.523**, **0.487** | 0.533, 0.491 | 0.531, 0.492 |
>
>
>
> ### **2. Larger-scale Experiments**
>
> To further address concerns about efficiency and performance on large-scale datasets, we conducted experiments on two large-scale time series datasets: XXLTraffic [1] and XTraffic [2]. The long-period dataset XXLTraffic includes 12 years of traffic flow data from 2012 to 2024, while the large-scale dataset XTraffic contains traffic data recorded since 2023 across 16,972 feature channels (nodes), showcasing a significantly larger number of channels.
>
> Due to time constraints, we compared the models you are concerned with (TimeKAN, DLinear, and the current SOTA time series model DUET). The input length was uniformly set to 512, and the output length to 720. The performance results and training efficiency comparisons on two datasets are as follows:
>
> | XXLTraffic | MSE   | MAE   | Memory  | Training Time|
> |----------|-------|-------|---------|------------|
> | MoFo (Ours)   | **0.066**  | **0.145**  | 1,486 MB | 149.1 s/epoch |
> | DUET   | 0.072  | 0.158  | 5,728 MB  | 296.7 s/epoch |
> | TimeKAN | 0.080  | 0.179  | 22,272 MB | 1008.6 s/epoch |
> | DLinear | 0.074  | 0.163  | 792 MB  | 84.4 s/epoch   |
>
> | XTraffic | MSE   | MAE   | Memory   | Epoch Time   |
> |--------|-------|-------|----------|------------|
> | MoFo (Ours)   |**0.048** | **0.090** |32,182 MB  | 765.1 s/epoch
> | DUET   |   -   |    -      |     Out-of-Memory    |  - |
> | TimeKAN |   -   |     -       |     Out-of-Memory  | -    |
> | DLinear | 0.062  | 0.107  | 14,548 MB | 552.15 s/epoch |
>
> In larger-scale scenarios, MoFo demonstrates the ability to achieve the highest performance while maintaining competitive efficiency. Specifically, for XTraffic dataset with a large number of nodes (channels), TimeKAN and DUBT are too complex to run. Compared to DLinear, MoFo achieves a 22.58% performance improvement on the large-scale dataset. Furthermore, for the long time-span dataset, MoFo achieves competitive performance while maintaining high efficiency.
>
> Ref:
>
> [1] Yin D, Xue H, Prabowo A, et al. Xxltraffic: Expanding and extremely long traffic dataset for ultra-dynamic forecasting challenges[J]. arXiv e-prints, 2024: arXiv: 2406.12693.
>
> [2] Gou X, Li Z, Lan T, et al. Xtraffic: A dataset where traffic meets incidents with explainability and more[J]. arXiv preprint arXiv:2407.11477, 2024.

---

> > ### Comment · Reviewer_7GM5 · 2025-08-06
> >
> > I have no further concerns. I suggest that the authors include the above points from the rebuttal in the revised paper for clarity. I am raising my score to 4.

---

> > > ### Author Response · Authors · 2025-08-06
> > >
> > > Thank you very much for your time and feedback. We promise to incorporate our discussion into the new version. Once again, we sincerely appreciate your encouraging evaluation.

---

### Official Review · Reviewer_EWUL · 2025-07-03

**Clarity:** 3
**Significance:** 3
**Originality:** 2
**Rating:** 4
**Confidence:** 3

**Summary:**

The paper introduces MoFo, a Transformer-based long-term time-series forecasting model comprising
1. Period-Structured Patching (PSP): reshapes an input length-$T$ series into $P$ rows of phase-aligned "patches", reducing the number of tokens processed;
2. Period-Aware Modulator (PAM): multiplies attention logits by a learnable sigmoid mask of phase distances, theoretically shown to approximate a hard phase gate.
The authors claim significant memory and runtime savings, as well as competitive accuracy, on eight periodic benchmarks, and provide a theoretical bound on the mask approximation error.

**Questions:**

How does accuracy change if the supplied $P$ is perturbed by ±5–15 % or if multiple periods (e.g., 24 h & 7 d) coexist?

**Ethical Concerns:**

["NO or VERY MINOR ethics concerns only"]

**Final Justification:**

The authors have addressed most of my other concerns. I suggest that the authors provide detailed explanations from the rebuttal in the revised paper for clarity. I have raised my score to 4.

**Limitations:**

Yes

**Quality:**

2

**Strengths And Weaknesses:**

Strengths
S1) PSP and PAM are easy to implement and interpret; the paper is mostly well written.
S2) On datasets with a single, stable period, MoFo yields substantial token reduction and lower memory/FLOPs.
S3) The paper provides a concise proof that the sigmoid mask converges to a binary phase filter.

Weaknesses
W1) The proposed method relies on the assumption of a period length $P$. However, real-world time series often exhibit complex, time-varying periodic patterns. For example, in Figure 1(a), the "Successive Patching" clearly shows multiple peaks representing mixed periodicities. Why is it reasonable to treat these as belonging to the same period? When dealing with multiple samples from different time periods, how can we ensure accurate and meaningful period selection? When data faces mixed periodicities and time-varying periods (e.g., summer vs. winter electricity consumption patterns), can MoFo effectively handle such complex periodic variations?

---

> ### Author Rebuttal · Authors · 2025-07-31
>
> We sincerely thank you for your precious time and insightful comments. We will now address your concerns point by point below.
>
> ---
> ## **W1.**
>
> **W1.1. Confusion about period**
>
>
> In Figure 1(a), the “Successive Patching” does not represent the coexistence of mixed periods; rather, it includes multiple peaks, which are dynamic components within a single period and together constitute a complete period.
>
> **We cannot simply assume that a single peak corresponds to one complete period.** For instance, in traffic data, there are distinct morning and evening peak periods; however, the periodicity of traffic is typically considered to be one full day rather than 12 hours. To accurately determine the period length, we employ the autocorrelation function (ACF). As shown in Figure 1(b), the autocorrelation coefficients of the sequence in Figure 1(a) indicate that a truly complete period often contains multiple peaks.
>
> **W1.2. Accurate period selection**
>
> As stated in Appendix Section A.2 of the manuscript, we utilize three steps to accurately determine periodicity.
>
> Firstly, we identify potential period lengths of repeating patterns within the sequence using the autocorrelation function (ACF). The ACF is a widely used tool in the field of time series analysis for examining periodic patterns [1-3].
>
> Secondly, we verify the existence of these periods through statistical significance testing, such as utilizing a 95% confidence interval, to eliminate any noise interference.
>
> Finally, we also conduct hyperparameter experiments to assess the sensitivity of model performance with respect to different period lengths.
>
> **W1.3. Time-varying periods**
>
> The electricity dataset (ETT) used in our experiments aligns perfectly with your requirements. The ETT dataset records electricity consumption from two counties in China over a continuous period from July 2016 to July 2018, thus encompassing both winter and summer patterns. However, it is worth noting that the periodic lengths for the two seasons do not differ significantly. For your convenience, we have included the relevant performance results below.
>
> | Dataset |Method  | MoFo         | DUET         | DLinear      |
> |:--------:|:--------:|:--------------:|:--------------:|:--------------:|
> |          | Prediction Window    |       MSE, MAE      |     MSE, MAE        |     MSE, MAE        |
> | ETTh1  | 96     | 0.360, 0.389 | **0.352**, **0.384** | 0.379, 0.403 |
> |        | 192    | **0.397**, 0.413 | 0.398, **0.409** | 0.408, 0.419 |
> |        | 336    | **0.407**, **0.424** | 0.414, 0.426 | 0.440, 0.440 |
> |        | 720    | 0.447, **0.454** | **0.429**, 0.455 | 0.471, 0.493 |
> | ETTm1  | 96     | 0.286, 0.335 | **0.279**, **0.333** | 0.300, 0.345 |
> |        | 192    | **0.320**, 0.363 | **0.320**, **0.358** | 0.336, 0.366 |
> |        | 336    | **0.347**, 0.382 | 0.348, **0.377** | 0.367, 0.386 |
> |        | 720    | **0.388**, 0.411 | 0.405, **0.408** | 0.419, 0.416 |
>
> Furthermore, to address your concerns more comprehensively, we have created a synthetic dataset (Mixed-ETT) by systematically combining ETTh1 (with a period of 24) and ETTm1 (with a period of 96). We divided each dataset into four equal parts along the time axis. Then, we alternately sampled each segment to construct the new dataset Mixed-ETT, represented as {ETTh1[:$\frac{1}{4}$], ETTm1[$\frac{1}{4}$:$\frac{1}{2}$], ETTh1[$\frac{1}{2}$:$\frac{3}{4}$], ETTm1[$\frac{3}{4}$:1]}. Consequently, the Mixed-ETT dataset displays a mixed periodic length of (24, 96, 24, 96). We compare the performance of state-of-the-art models, DUET and DLinear, as shown below.
>
> |  Dataset    | Method| MoFo         | Duet         | DLinear     |
> |:------------:|:--------:|:--------------:|:-------------:|:------------:|
> |          | Prediction Window    |       MSE, MAE      |     MSE, MAE        |     MSE, MAE        |
> | Mixed-ETT | 96     | **0.178**, **0.223** | 0.186, 0.229 | 0.194, 0.232 |
> |            | 192    | 0.185, **0.232** | **0.184**, 0.241 | 0.190, 0.247 |
> |            | 336    | **0.180,** **0.237** | 0.187, 0.247 | 0.196, 0.256 |
> |            | 720    | **0.191**, **0.253** | 0.218, 0.278 | 0.219, 0.282 |
>
> Even under time-varying periods, MoFo retains the competitive performance, underscoring the soundness of its architectural design and its robust capability to capture dynamic temporal patterns.
>
> ---
>
> ## **Q1.**
>
> **Q1.1. Performance under perturbed period setting**
>
> To resolve any remaining concerns, we introduced four variants:
>
> - **"+5 %"** and **"+15 %"**: the period setting is increased by 5 % and 15 %, respectively.
> - **"–5 %"** and **"–15 %"**: the period setting is decreased by 5 % and 15 %, respectively.
>
> We evaluated these perturbations on three datasets with distinct period lengths; the quantitative results are reported below.
>
> | Dataset | Method   | MoFo | + 5% | + 15% | - 5% | - 15% |
> |:---------:|:---------:|:---------:|:---------:|:---------:|:---------:|:-----------------:|
> |          | Prediction Window   |       MSE, MAE      |     MSE, MAE        |     MSE, MAE        |     MSE, MAE        |     MSE, MAE        |
> | ETTm1   | 96     | **0.286**, **0.335**     | 0.292, 0.335     | 0.297, 0.338     | 0.291, 0.337     | 0.297, 0.341     |
> |         | 192    | **0.320**, **0.363**     | 0.329, 0.368     | 0.336, 0.370     | 0.333, 0.369     | 0.344, 0.386     |
> |         | 336    | **0.347**, **0.382**     | 0.357, 0.388     | 0.372, 0.391     | 0.354, 0.385     | 0.376, 0.402     |
> |         | 720    | **0.388**, **0.411**     | 0.406, 0.420     | 0.403, 0.416     | 0.402, 0.415     | 0.411, 0.420     |
> | Weather | 96     | **0.141**, **0.186**     | 0.143, 0.192     | 0.155, 0.206     | 0.157, 0.221     | 0.146, 0.188     |
> |         | 192    | **0.186**, **0.230**     | 0.196, 0.236     | 0.202, 0.252     | 0.199, 0.240     | 0.201, 0.242     |
> |         | 336    | **0.233**, **0.272**     | 0.238, 0.277     | 0.245, 0.286     | 0.235, 0.280     | 0.243, 0.286     |
> |         | 720    | **0.312**, **0.331**     | 0.335, 0.358     | 0.323, 0.358     | 0.332, 0.352     | 0.335, 0.355     |
> | Traffic | 96     | **0.362**, **0.247**     | 0.388, 0.266     | 0.395, 0.265     | 0.383, 0.255     | 0.382, 0.265     |
> |         | 192    | **0.379**, **0.254**     | 0.397, 0.275     | 0.408, 0.287     | 0.398, 0.284     | 0.409, 0.288     |
> |         | 336    | **0.390**, **0.258**     | 0.395, 0.260     | 0.405, 0.275     | 0.395, 0.266     | 0.417, 0.294     |
> |         | 720    | **0.424**, **0.281**     | 0.424, 0.281     | 0.443, 0.298     | 0.437, 0.283     | 0.443, 0.303     |
>
> ---
>
> **Q1.2. Performance under multiple periods setting**
>
> We utilized the traffic dataset, which includes the multiple time intervals you mentioned: 1 day and 7 days. In practice, when dealing with multi-period data, we typically set the shorter period as our baseline configuration. Our experimental results were very impressive, as shown in Table 2 in the manuscript.
>
> To address your concern regarding the performance when integrating multiple MoFo models to handle different periods, we introduced three models. One original model (MoFo-1) excels in sequences with a period length of 24 hours (1 day), another (MoFo-7) is designed for a weekly period of 7 days, and the final ensemble model (Mix-MoFo) learns and predicts in parallel using the two MoFo models, combining their outputs to produce the final prediction. Below are the results from our experiments on the Traffic dataset.
>
> The experimental results indicate that using the shorter period as the configuration allows MoFo to capture more precise periodic patterns, leading to enhanced performance.
>
> | |  | MoFo-1|Mix-MoFo| MoFo-7|
> |--------|------|----------------|------|------|
> |          | Prediction Window   |       MSE, MAE      |     MSE, MAE        |     MSE, MAE        |
> | Traffic | 96    | **0.362**, **0.247** | 0.365, 0.254 | 0.381, 0.257 |
> |        | 192   | **0.379**, **0.254** | 0.380, 0.255 | 0.394, 0.262 |
> |        | 336   | **0.390**, **0.258** | 0.395, 0.263 | 0.409, 0.272 |
> |        | 720   | **0.424**, **0.281** | 0.430, 0.285 | 0.451, 0.289 |
>
> Experimental results demonstrate that our straightforward MoFo implementation achieved strong performance. When dealing with multi-period time series, we set the smallest period length as our baseline configuration. The “Mix-MoFo” variant added assumptions regarding additional periods, which increased the risk of overfitting and consequently led to a decline in performance.
>
> Ref:
>
> [1] Wu H, Xu J, Wang J, et al. Autoformer: Decomposition transformers with auto-correlation for long-term series forecasting[J]. Advances in neural information processing systems, 2021, 34: 22419-22430.
>
> [2] Lin S, Lin W, Hu X, et al. Cyclenet: Enhancing time series forecasting through modeling periodic patterns[J]. Advances in Neural Information Processing Systems, 2024, 37: 106315-106345.
>
> [3] Lin S, Lin W, Wu W, et al. SparseTSF: Modeling Long-term Time Series Forecasting with* 1k* Parameters[C]//International Conference on Machine Learning. PMLR, 2024: 30211-30226.

---

> > ### Comment · Reviewer_EWUL · 2025-08-09
> >
> > Thank you for your response. I have read the response and most of my concerns have been addressed. I have improved the score. I would like to suggest adding more detailed explanations and experimental results from the rebuttal in the revised paper.

---

> ### Comment · Area_Chair_P8GZ · 2025-08-05
> **Follow-up on author rebuttal**
>
> Hi Reviewer EWUL,
>
> The authors have submitted their rebuttal. Do you have any further questions or comments based on their response?
>
> Best regards,
> The AC

---

> ### Author Response · Authors · 2025-08-06
> **Seeking the Reviewer’s Valuable Feedback**
>
> Dear Reviewer EWUL,
>
> Thank you very much for your thorough review. If you have any further questions or suggestions, we would be delighted to discuss them with you.
>
> To make our response clearer, we have summarized it as follows:
>
> (1) We would like to clarify that the selection of the period length is determined using the autocorrelation function with Bartlett's Test under the Null Hypothesis, a widely adopted method for periodic analysis in time series data. Further details can be found in Appendix A.2 of the paper.
>
>
> (2) We designed experiments to evaluate the model's performance in dynamic-, mixed-, and multi-period scenarios. The highly competitive results demonstrate the outstanding performance of our model under these complex conditions.
>
> Additionally, we are happy to share the latest progress of our experiments.
>
> In our response to Reviewer 7GM5, **we utilized a 12-year-long time series dataset, XXLTraffic, which potentially contains complex periodic scenarios over such an extended time span**. On this dataset, our model also achieved competitive performance. For convenience, we have reproduced the performance comparison below.
>
> | XXLTraffic | MoFo (Ours) | DUET | TimeKAN | DLinear |
> |:-:|:-:|:-:|:-:|:-:|
> | MSE        | **0.066**        | 0.072 | 0.080   | 0.074   |
> | MAE        | **0.145**       | 0.158 | 0.179   | 0.163   |
>
> Thank you again for your professional and valuable review! Your feedback can significantly enhance the quality of our paper, and we sincerely appreciate it.
>
> Sincerely,
>
> The authors of the paper.

---

> ### Author Response · Authors · 2025-08-09
> **Thanks**
>
> Dear Reviewer EWUL,
>
> We are deeply grateful for your thoughtful and constructive feedback, as well as the time and effort you have dedicated to reviewing our manuscript. Your insightful suggestions have been invaluable in guiding us to enhance the quality of our research.
>
> Once again, we sincerely thank you for your support and expertise, and we truly appreciate your valuable contribution to this work.
>
> With warm regards,
>
> The Authors of Submission 2224

---

### Decision · Program_Chairs · 2025-09-17

**Decision:**

Accept (poster)

**Comment:**

This paper proposes a modulated method for period modeling in long-term time series forecasting. All reviewers acknowledged the strong motivation, clear presentation, concise model design, and (after rebuttal) comprehensive evaluation. Following the rebuttal, two reviewers increased their ratings to positive scores, and the overall consensus was in favor of acceptance.

I strongly encourage the authors to revise the paper in line with the reviewers’ comments and to release their code to the community to enhance reproducibility.

Recommendation: Accept as poster.